# Optimal Information Retention for Time-Series Explanations

**Jinghang Yue**[1]  **Jing Wang**[1,2]  **Lu Zhang**[1]  **Shuo Zhang**[1]  **Da Li**[1]  **Zhaoyang Ma**[1]  **Youfang Lin**[1,3]

## Abstract

Explaining deep models for time-series data is crucial for identifying key patterns in sensitive domains, such as healthcare and finance. However, due to the lack of a unified optimization criterion, existing explanation methods often suffer from redundancy and incompleteness, where irrelevant patterns are included or key patterns are missed in explanations. To address this challenge, we propose the Optimal Information Retention Principle, where conditional mutual information defines minimizing redundancy and maximizing completeness as optimization objectives. We then derive the corresponding objective function theoretically. As a practical framework, we introduce an explanation framework ORTE, learning a binary mask to eliminate redundant information while mining temporal patterns of explanations. We decouple the discrete mapping process to ensure the stability of gradient propagation, while employing contrastive learning to achieve precise filtering of explanatory patterns through the mask, thereby realizing a trade-off between low redundancy and high completeness. Extensive quantitative and qualitative experiments on synthetic and real-world datasets demonstrate that the proposed principle significantly improves the accuracy and completeness of explanations compared to baseline methods. The code is available at https://github.com/moon2yue/ORTE_public.

## 1. Introduction

The rapid advancements in deep learning have greatly improved the analysis of time-series data, excelling particularly in capturing intricate temporal patterns and managing high-throughput datasets. However, their inherently multi-layered and nonlinear architecture often renders them black-box, prompting significant concerns regarding interpretability and reliability in sensitive domains, such as healthcare (Alqaraawi et al., 2020) and finance (Mokhtari et al., 2019). For instance, explaining the temporal patterns and complex dynamics underpinning deep model predictions is essential for understanding the physiological representation of diseases and enabling effective early warning systems (Di Martino & Delmastro, 2023).

Research on improving the interpretability of deep models for time series signals primarily focuses on local explanations, aiming to identify critical temporal patterns by locating significant positions in time series signals. These approaches have attracted considerable interest due to their ability to provide intuitive, data-driven insights, highlighting which features are essential or irrelevant for specific downstream tasks. Some existing local explanation methods are adaptations of approaches originally developed for computer vision (CV) and natural language processing (NLP). However, Ismail et al. (2020) illustrates that saliency-based methods, such as Integrated Gradients (Baehrens et al., 2010), Deep SHAP (Scott et al., 2017), and Feature Occlusion (Zeiler & Fergus, 2014), fail to reliably and accurately identify the temporal variation in feature importance within time series data. Others are tailored specifically for time-series data. For example, Dynamask (Crabbé & Van Der Schaar, 2021) incorporates temporal dependencies by learning the effects of perturbation operators, enabling it to generate instance-specific importance scores for each feature at every time step. TIMEX (Queen et al., 2024) trains a surrogate model to replicate the predictive behavior of a black-box model, enabling the identification of important positions and temporal patterns. However, these heuristic methods lack a unified optimization framework and often overlook the requirements for explanation completeness or low redundancy, posing challenges for identifying critical temporal patterns. As illustrated in Figure 1, the mixing of redundant patterns makes critical patterns unable to be highlighted, while incompleteness may cause missing patterns or even

---

[1]School of Computer Science and Technology, Beijing Jiaotong University, Beijing, China [2] Key Laboratory of Big Data & Artificial Intelligence in Transportation (Beijing Jiaotong University), Ministry of Education, Beijing, China [3]Beijing Key Laboratory of Traffic Data Mining and Embodied Intelligence, Beijing, China. Correspondence to: Jing Wang <wj@bjtu.edu.cn>.

*Proceedings of the 42nd International Conference on Machine Learning*, Vancouver, Canada. PMLR 267, 2025. Copyright 2025 by the author(s).

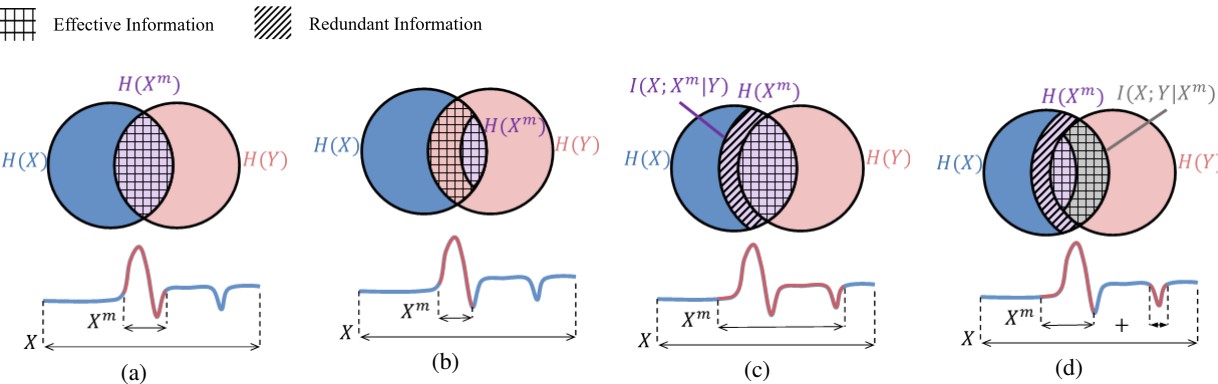

*Figure 1.* The Venn Diagram of information retention. $I(X;Y)$ is defined as task-relevant effective information. (a) Ideal case: $X^m$ only contains the same information as $I(X;Y)$. (b) Incompleteness case: The information of $X^m$ is a proper subset of $I(X;Y)$. (c) Redundant case: The information of $X^m$ contains not only $I(X;Y)$, but also other information irrelevant to $Y$. (d) Mixed case: A mixture of incompleteness and redundant cases.

pointing to wrong patterns.

Information theory provides a novel perspective for explaining the complex dynamics of deep models. Representative approaches, such as the Information Bottleneck (IB) (Tishby & Zaslavsky, 2015), focus on minimizing the mutual information $I(X;Z)$ between the input $X$ and the learned representation $Z$, while maximizing the mutual information $I(Y;Z)$ between $Z$ and the labels $Y$. The IB framework structurally characterizes the learning process as a trade-off between compact representations and informational utility. TIMEX++ (Liu et al., 2024) highlights that many existing interpretability metrics for time-series data are prone to trivial solutions and issues arising from distribution shifts. To address these challenges, an optimization objective for interpretability is derived from IB. Contrastive learning, on the other hand, centers on minimizing the distance between positive pairs in the embedding space while maximizing the distance between negative pairs (Dosovitskiy et al., 2014; Khosla et al., 2020; Tian et al., 2020). For instance, training models to maximize the mutual information of positive pairs and minimize that of negative pairs aligns well with this framework (He et al., 2020). Rate reduction focuses on compressing intra-class informational bits, ensuring that representations of different classes occupy highly uncorrelated linear subspaces (Yu et al., 2020; Chan et al., 2022). CRATE (Yu et al., 2023) interprets the Transformer model as a multi-layer stack of information compression and discretization processes, and introduces a simplified, white-box version of the Transformer model. While these methods provide insights into the learning process or objectives of deep models from various perspectives, they are not designed to directly guide the optimization of local explanations. Due to the dynamic nature of time series, redundant or incomplete local segments may represent different temporal patterns. Additionally, the heterogeneity of data poses challenges

for heuristic methods in generalizing to time series signals. Therefore, uniform optimization principles and practical frameworks for time series signals are needed to be directly applied to improve the local interpretation of deep models.

To address the above challenges, we propose the **Optimal Information Retention Principle** grounded in information theory for precisely identifying explanatory temporal patterns within time series. This principle encompasses three criteria to guide the optimization of local explanations: *i*) Semantic Information Retention, which aligns the prediction distribution of the interpreter with the black-box model to ensure fidelity of explanations; *ii*) Minimum Redundant Information Retention, which minimizes the mutual information between the input and the explanation under conditions unrelated to downstream tasks, thereby avoiding the inclusion of redundant information; *iii*) Maximum Effective Information Retention, which ensures that after removing the explanatory temporal patterns, the mutual information between the input and the label is minimized. This implies that the input loses its predictive capability, thereby guaranteeing the completeness of the explanation.

Based on the Optimal Information Retention Principle, we have derived objective functions for distribution alignment, redundancy elimination, and information completeness, respectively. Furthermore, we propose an **O**ptimal Information **R**etention for **T**ime-Series **E**xplanations (ORTE) as a practical framework. Specifically, we first construct a parametric network to learn a binary mask for filtering redundant information from the input. We decouple the forward mapping and backward gradient propagation processes to mitigate the issue of gradient instability in binary mask learning. Then, we utilize the binary mask to construct positive and negative sample pairs and employ contrastive learning to achieve a trade-off between low redundancy and

high completeness in the explanations. Additionally, we align the prediction distribution of the explanations with the black-box model to ensure the fidelity of the explanations. Our contributions are summarized as follows:

- We propose the Optimal Information Retention Principle grounded in information theory to address the issues of redundancy and incompleteness in time series interpretation, and theoretically derive the corresponding objective functions.

- We propose ORTE as a practical framework, which achieves a trade-off between low redundancy and high completeness in explanations by decoupling discrete mapping, contrastive learning, and distribution alignment.

- We achieve state-of-the-art performance on eight synthetic and real-world time series datasets compared to the latest competitive baselines and validate the utility of each component.

## 2. Optimal Information Retention Principle

In this section, we first formally describe the formulation of time-series explanations based on classification task. Then, we present a unified optimization principle for time-series explanations, namely the principle of optimal information retention. This principle comprises three criteria: semantic information retention, minimal redundant information retention, and maximal effective information retention. Based on these criteria, we theoretically derive objective functions for optimizing time-series explanations.

### 2.1. Problem Formulation

**Time Series Classification.** Given a time series dataset $\mathcal{D} = (X, Y) = \{(x_i, y_i)\}_1^N$ containing $N$ pairs, $X$ are multivariate or univariate time series instances, and $Y$ are the corresponding labels. Each time series instance $x_i \in \mathbb{R}^{T \times D}$ is composed of vectors collected by one (e.g., $D = 1$) or multiple sensors (e.g., $D > 1$), where $T$ denotes the length of time steps and $D$ represents the number of sensors. Each label $y_i$ belongs to one class of $\{1, 2, \ldots, C\}$. A pre-trained deep neural network (DNN) $f$ is trained to classify the input instance $x_i$ into the maximum probability class, e.g., $\hat{y}_i = argmax(f(x_i)) \in C$.

**Local explanation for time series.** In this paper, we aim to propose a mask generator $g(f, x)$ to learn a binary mask matrix $M \in \{0, 1\}^{T \times D}$, which can retain the important pieces of $x$, e.g., $X^m = X \odot M$, where $\odot$ is an element-wise multiplication. In other words, $X^m$ are considered more important than $X \odot (1 - M)$ for the classification task. Naturally, the ideal sub-features $X^m$ are expected to retain the most relevant information to the downstream task.

The probability distribution of $M$ can also be regarded as a salient map (attribution map or importance map).

### 2.2. Optimal Information Retention

For a task $\mathcal{T}$ whose goal is to predict label $Y$ from the input $X$, the optimal sub-feature $X^m \in X$ should contain minimal irrelevant and maximum effective information to the mapping $f : X \to Y$. In order to introduce the optimal explanations $X^m$, we first investigate the possible cases of information retention within $X^m$ as shown in Figure 1. Ideally, the optimal explanation $X^m$ exclusively and entirely contains the task-relevant information within $X$, denoted as Figure 1(a). Figure 1(b) shows the incompleteness information case, where $X^m$ is a subset of $I(X; Y)$, failing to fully explain the predictive behavior of $f(X)$. This may manifest in the explanation as only highlighting partial fragments of the effective series. On the other hand, a redundant information case occurs when $X^m$ not only includes the necessary explanatory information but also incorporates irrelevant information, as shown in Figure 1(c). This may lead to the mixing of irrelevant patterns, causing critical temporal patterns to fail to be highlighted. Figure 1(d) involves a combination of incompleteness and redundant information within a single context, which may be a more challenging case. Both the inclusion of redundant information and the absence of effective information can alter critical temporal patterns, which is a key challenge in time-series explanations. Furthermore, explanations should remain faithful to the pre-trained black-box model. To this end, we propose the optimal information retention principle containing three criteria: Semantic Information Retention, Minimum Redundant Information Retention, and Maximum Effective Information Retention, as outlined below.

**Criteria 2.1.** (Semantic Information Retention) To faithfully reflect the predictive behavior $f(X)$, $X^m$ should perform similar task-specific semantic consistency, as follows:

$$\min \mathcal{L}_{\text{JS}}\left(f(X), f\left(X^m\right)\right), \quad (1)$$

where $\mathcal{L}_{\text{JS}}$ denotes the Jensen-Shannon divergence.

Criteria 2.1 ensures the fidelity of the explanation by aligning the distributions of $f(X)$ and $f(X^m)$. This avoids spurious correlations between the interpreter and the pre-trained model, meaning that the identified key temporal patterns are indeed the basis for the black-box model's predictions. Furthermore, Semantic Information Retention also aligns with the advantages of prediction consistency emphasized in TIMEX (Queen et al., 2024) and the *signal issue* addressed in TIMEX++ (Liu et al., 2024).

**Criteria 2.2.** (Minimum Redundant Information Retention) The optimal sub-features $X^m$ should only contain the mutual information shared by $I(X; Y)$, and include as little

information of $X$ no relation to $Y$, as follows:

$$\min I\left(X; X^m \mid Y\right). \tag{2}$$

Although Criteria 2.2 provides an ideal direction for minimizing redundant information, computing the conditional mutual information remains difficult for:

$$I\left(X; X^m \mid Y\right) = \mathbb{E}_{p(x,y)}\left[\mathbb{E}_{p(x^m|x,y)}\left[\log\frac{p\left(x^m \mid x,y\right)}{p\left(x^m \mid y\right)}\right]\right], \tag{3}$$

where $p\left(x^m \mid x,y\right)$ and $p\left(x^m \mid y\right)$ are intractable in DNN. Considering $X^m = X \odot M$, $I\left(X; M \mid Y\right)$ is served as the simplified form of $I\left(X; X^m \mid Y\right)$. Furthermore, by relating it to the Kullback-Leibler divergence, there are

$$I\left(X; M \mid Y\right) = \mathbb{E}_{p(x,y)}\left[D_{\mathrm{KL}}(p(m \mid x,y)\|p(m \mid y))\right], \tag{4}$$

where $I\left(X; X^m \mid Y\right) \propto D_{\mathrm{KL}}(p(m \mid x,y)\|p(m \mid y))$. Referring to the variational approaches (Miao et al., 2022; Queen et al., 2024), $M$ can be sampled following Bernoulli distribution at each time step, *i.e.*, $p(m \mid x) = \prod_{t,d} \mathrm{Bern}(p_{t,d})$, where $p_{t,d}$ denotes the probability that the value of $x$ at time $t$ and dimension $d$ is retained. Assuming the desired distribution of $p$ is $p(m) = \prod_{t,d} \mathrm{Bern}(r)$, the derived objective function can be expressed as:

$$\mathcal{L}_{\mathrm{mask}} = \mathbb{E}[D_{\mathrm{KL}}(p(m \mid x)\|q(m))]$$
$$= \sum_{t,d}[p_{t,d} \log\frac{p_{t,d}}{r} + (1 - p_{t,d})\log\frac{1 - p_{t,d}}{1 - r}], \tag{5}$$

where $r$ as the desired probability of $q(m = 1)$ is a hyperparameter in practice. The detailed proof of (5) is provided in Appendix B.1.

Intuitively, the optimal $X^m$ should contain all the information relevant to $Y$. Assuming $X^m$ is removed, $I\left(X; Y \mid X^m\right) \to 0$ indicates that $H(X \mid X^m)$ completely loses the information required to predict $Y$. To this end, the formal definition of maximum effective information retention is as follows.

**Criteria 2.3.** (Maximum Effective Information Retention) The optimal retention of sub-features $X^m$ should encompass as much task-relevant information from $X$ as possible, minimizing the risk of information omission, as follows:

$$\min I\left(X; Y \mid X^m\right). \tag{6}$$

We pad the masked parts of $X^m$ to obtain $\hat{X}$, ensuring consistency in dimensions while avoiding the Out-of-Distribution (OOD). Furthermore, (6) can be decomposed as:

$$I\left(X; Y \mid X^m\right) \Rightarrow$$
$$I\left(X; Y \mid \hat{X}\right) = I\left(X; Y, \hat{X}\right) - I\left(X; \hat{X}\right). \tag{7}$$

To minimize $I\left(X; Y \mid X^m\right)$, we minimize the first term $I\left(X; Y, \hat{X}\right)$, while maximize the second term $I(X; \hat{X})$.

For the first term, when $\hat{X}$ serves as the input generated by $X^m$, the mutual information $I(X; \hat{X})$ should ideally include only the information contained in $I(X; Y)$. Otherwise, redundant information may be introduced. Assuming $\hat{X}$ has no overlap with the portions of $H(X \mid Y)$, then $I\left(X; Y, \hat{X}\right) = I\left(X; Y\right)$, which is a constant independent of $\hat{X}$.

For the second term, since $X$ and $\hat{X}$ share effective information in $X^m$, they can naturally regarded as the anchor samples and the positive samples, respectively, in contrastive learning. Therefore, we employ contrastive loss to maximize the mutual information between $X$ and $\hat{X}$:

$$\mathcal{L}_{\mathrm{ct}} = -\frac{1}{N}\sum_{i=1}^{N}\{\log[\exp\left(f\left(x_i\right) \cdot f\left(\hat{x}_i\right)\right)]$$
$$- \log[\exp\left(f\left(x_i\right) \cdot f\left(\hat{x}_i\right)\right) + \sum_{j=1}^{K}\exp\left(f\left(x_i\right) \cdot f\left(\hat{x}_j^-\right)\right)]\}, \tag{8}$$

where $\hat{x}_j^-$ denotes the negative sample, which can be generated by $h(X - X^M)$, i.e., Gaussian noise imputation. The detailed proof of (8) is provided in Appendix B.2.

## 3. ORTE Method

We now present the **O**ptimal **I**nformation **R**etention for **T**ime-Series **E**xplanations (ORTE), whose overview framework is illustrated in Figure 2. Based on the analysis in the previous section, ORTE describes a practical framework for time-series explanations from the optimal information retention principle. ORTE includes three main blocks like Adaptive Mask Generator (AMG), Contrastive Explanation Separator (CES), and Predictive Distribution Aligner (PDA). Specifically, AMG learns a 0-1 occlusion matrix to mask redundant information so that achieves the *Minimum Redundant Information Retention*. CES employs contrastive learning to trade off low redundancy and high completeness in the explanations, thereby achieving the *Maximum Effective Information Retention*. PDA aligns the prediction distribution of the explanations with the black-box model to ensure the *Semantic Information Retention*.

### 3.1. Adaptive Mask Generator

According to the *Minimum Redundant Information Retention*, the binary mask $M \in \{0,1\}^{T \times D}$ reveals the interpretability of the time series, i.e., important segments are assigned $\{1\}$ to be retained. The explanation generator is performed through two steps to learn $M$: Probabilistic Generation Process and Discrete Mapping Process.

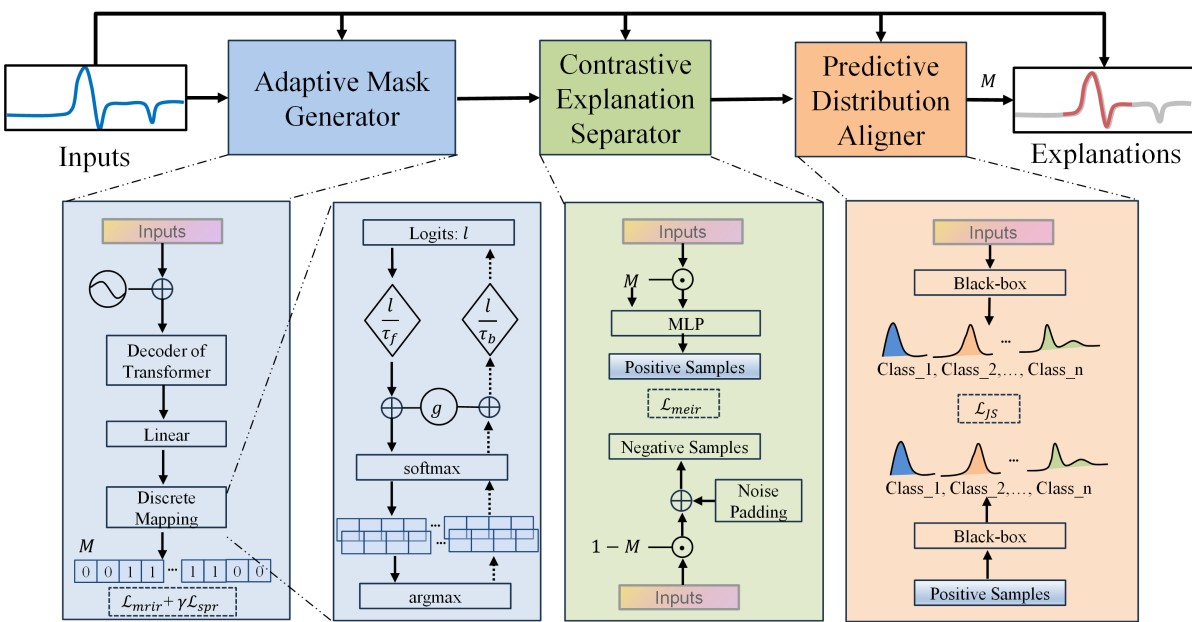

*Figure 2.* Overview of ORTE method including three main blocks: Adaptive Mask Generator (AMG), Contrastive Explanation Separator (CES), and Predictive Distribution Aligner (PDA).

**Probabilistic Generation Process.** The first process encodes the input $X$ as a probabilistic matrix $P(M \mid X)$, where $p_{t_1,d_1} > p_{t_2,d_2}$ indicates $x_{t_1,d_1}$ is more important than $x_{t_2,d_2}$. As discussed by Queen et al. (2024), we employ an autoregressive transformer decoder and softmax activations to output the probability of each temporal sensor pair. Additionally, to optimize the continuity of the distribution, we also introduce a continuous loss:

$$\mathcal{L}_{\text{con}} = \frac{1}{T \times D} \sum_{d=1}^{D} \sum_{t=1}^{T-1} \sqrt{(p_{t,d} - p_{t+1,d})^2}. \quad (9)$$

So the objective function for minimal redundant information retention can be expressed as $\mathcal{L}_{\text{mrir}} = \mathcal{L}_{\text{mask}} + \mathcal{L}_{\text{con}}$.
**Discrete Mapping Process.** Mapping $P$ from the continuous space to the discrete binary space $M$ suffers from non-differentiable gradients. Queen et al. (2024) introduces the Straight-Through Gumbel-Softmax (ST-GS) estimator (Jang et al., 2016) to bridge the gap between the forward propagation mapping discrete distribution and the backward derivation being non-differentiable. Let $l$ denotes the unnormalized logits of encoder outputs, the forward pass can be summarized as:

$$\textit{forward: } p = \text{softmax}\left(\frac{l}{\tau} + g\right)$$
$$m = \arg\max(p), \quad (10)$$

where $\tau$ represents the temperature parameter that controls the entropy of $p$, and $g$ is randomly sampled noise from Gumbel (0,1). Giving $\mathcal{L}$ as the loss function, the backward

pass is:

$$\textit{backward: } \frac{\partial \mathcal{L}}{\partial l} = \frac{\partial \mathcal{L}}{\partial m} \frac{\partial m}{\partial p} \frac{\partial p}{\partial l} \approx \frac{\partial \mathcal{L}}{\partial m} \frac{\partial p}{\partial l}, \quad (11)$$

where Straight-Through Estimator (STE) (Bengio et al., 2013) is applied to allow gradients to propagate through the logits, such as $\frac{\partial m}{\partial p} \approx 1$. Although ST-GS solves the problem of non-differentiable gradients in the discrete mapping process, it ignores the difference between forward and backward propagation. Specifically, the forward propagation requires the distribution of $p$ to be sharp with the training process so that extreme values are taken without significant bias. However, as discussed in Shah et al. (2024) the backpropagation process of the fixed gradient limits the ability to model the data distribution. We propose adaptive STE (*adapt*-STE) to alleviate the asymmetry of forward and backward propagation:

$$\tau_f = \text{clip}\left(\tau_f - \eta \cdot \sigma(l), \tau_{\min}, \tau_{\max}\right)$$
$$\tau_b = \text{clip}\left(\tau_f + \eta \cdot \sigma(l), \tau_{\min}, \tau_{\max}\right), \quad (12)$$

where $\tau_f$ and $\tau_b$ represent the temperature parameter of the forward propagation and backward propagation, respectively. $\text{clip}()$ restricts the upper and lower bound hyperparameters of $\tau_f$ and $\tau_b$ to be $\tau_{\max}$ and $\tau_{\min}$, and $\eta$ is the updating rate and $\sigma(l)$ is the standard deviation of $l$. A small $\tau_f$ ensures the sharpness of $p$ in the forward propagation, while a large $\tau_b$ ensures the faithfulness to $l$ in the backward propagation. The pseudocode of *adapt*-STE is summarized in Appendix C

## 3.2. Contrastive Explanation Separator

As discussed in Section 2.3, maximizing the task-relevant information shared by positive samples $\hat{X}$ with input $X$, and minimizing the retention of valid information by negative samples $\hat{X}^-$ is key to interpretation. To this end, we introduce how to generate positive and negative sample pairs. As discussed in TIMEX++ (Liu et al., 2024), it is essential to ensure the integrity of effective information while avoiding out-of-distribution (OOD) issues. Therefore, we adopt the same generation strategy:

$$\hat{X} = \text{MLP}(M, X), \tag{13}$$

where $\hat{X} \in \mathbb{R}^{T \times D}$. $\mathcal{L}_{\text{sim}} = mse(X, \hat{X})$ is employed to ensure generalization and avoid overfitting. For negative samples $\hat{X}^-$, we filter out effective information through reverse masking, while the masked portions are then filled with Gaussian noise:

$$\hat{X}^- = (1 - M) \odot X + M \odot pad, \tag{14}$$

where $pad \sim \mathcal{N}\left(\mu, \sigma^2\right)$. This effectively enhances the discrepancy between positive and negative samples, thereby facilitating a clearer distinction in the feature space. So the objective function for maximum effective information retention can be expressed as $\mathcal{L}_{\text{meir}} = \alpha \mathcal{L}_{\text{ct}} + \beta \mathcal{L}_{\text{sim}}$, where $\alpha$ and $\beta$ are hyperparameters.

## 3.3. Objective Function

**Predictive Distribution Aligner.** The explanation should be faithful to the predictions of the black-box model. For this purpose, we align the explained predictive distribution with the pre-trained black-box model predictive distribution. Referring to 1, the Jensen-Shannon divergence is minimized as the loss function to reduce the distribution difference.

The overall objective function for the learning process consists of four terms. First, $\mathcal{L}_{JS}$ aligns the explanation with the predictions of the black-box model, ensuring the retention of semantic information. Second, $\mathcal{L}_{mire}$ filters redundant information by learning a masking matrix. Third, $\mathcal{L}_{meir}$ decomposes the original data into positive and negative sample pairs, where contrastive learning is used to preserve effective information. Additionally, to enforce sparsity in the masking, we introduce $\mathcal{L}_{spr} = \|M\|_1$ as a regularization term. The overall objective function is as follows:

$$\mathcal{L} = \mathcal{L}_{\text{JS}} + \mathcal{L}_{\text{mrir}} + \mathcal{L}_{\text{meir}} + \gamma \mathcal{L}_{\text{spr}}, \tag{15}$$

where $\gamma$ is a hyperparameter.

## 4. Experiments

In this section, we conduct experiments to study the performance of our proposed ORTE method on four synthetic datasets and four real-world datasets. Without loss of generality, we employ the vanilla Transformer (Vaswani, 2017) as a black-box classifier. Our goal is to explain this black-box model. Here we first introduce the experimental settings including **Datasets**, **Metrics**, and **Comparison Methods**. Then the performance of the proposed method is compared with baseline methods and an ablation study is conducted to validate the effectiveness of individual modules. Finally, the ability of the optimal information retention principle to distinguish valid information from redundant information is experimentally verified. Refer to the Appendix D for more experimental details.

### 4.1. Experimental Settings

**Datasets.** We evaluate our method on four carefully designed synthetic datasets with ground-truth annotations: **FreqShapes**, **SeqComb-UV**, **SeqComb-MV**, and **LowVar** (Queen et al., 2024). These datasets, encompassing both univariate and multivariate time series, inherently contain complex temporal dynamics. The goal of the explanation is to identify key segments that represent critical patterns in the data. Additionally, we test on four real-world datasets: **ECG** - ECG arrhythmia detection (Moody & Mark, 2001); **PAM** - human activity recognition (Reiss & Stricker, 2012); **Epilepsy** - EEG seizure detection (Andrzejak et al., 2001); and **Boiler** - mechanical fault detection (Shohet et al., 2019). For the ECG dataset, we define the ground-truth explanation as the QRS intervals associated with arrhythmia detection. Makowski et al. (2021) provides standard reference points for the detection of R, P, and T waves in ECG signals.

**Metrics.** We employ two evaluation approaches. For *explanations with ground truth annotations*, we use the labeled explanations as a reference and quantify the results using Area Under Precision (AUP), Area Under Recall (AUR), and Area Under Precision-Recall Curve (AUPRC), which combines AUP and AUR. For all three metrics, higher values indicate better performance. For *explanations without ground truth annotations*, we follow an approach similar to Queen et al. (2024), where the bottom $p$-percentile of features identified by the explainer is masked, and the change in predictive Area Under Receiver Operating Characteristic Curve (AUROC) is measured.

**Comparison Methods.** We have compared the proposed method with seven explainability methods, as follows: Integrated gradients (**IG**) (Baehrens et al., 2010) is widely used for general interpretation; **Dynamask** (Crabbé & Van Der Schaar, 2021) and **WinIT** (Leung et al., 2021) are the time-series specific explanation methods; **CoRTX** (Chuang et al., 2023) is an explanation method based on contrastive learning; **SGT + Grad** (Ismail et al., 2021) is an *in-hoc* interpreter for time-series data; **TIMEX** (Queen et al., 2024) employs a white-box model as an interpreter; and **TIMEX++**

*Table 1.* Attribution explanation performance on univariate and multivariate synthetic datasets.

| METHOD | FREQSHAPES | | | SEQCOMB-UV | | |
|---|---|---|---|---|---|---|
| | AUPRC | AUP | AUR | AUPRC | AUP | AUR |
| IG | 0.7516±0.0032 | 0.6912±0.0028 | 0.5975±0.0020 | 0.5760±0.0022 | 0.8157±0.0023 | 0.2868±0.0023 |
| DYNAMASK | 0.2201±0.0013 | 0.2952±0.0037 | 0.5037±0.0015 | 0.4421±0.0016 | 0.8782±0.0039 | 0.1029±0.0007 |
| WINIT | 0.5071±0.0021 | 0.5546±0.0026 | 0.4557±0.0016 | 0.4568±0.0017 | 0.7872±0.0027 | 0.2253±0.0016 |
| CORTX | 0.6978±0.0156 | 0.4938±0.0004 | 0.3261±0.0012 | 0.5643±0.0024 | 0.8241±0.0025 | 0.1749±0.0007 |
| SGT + GRAD | 0.5312±0.0019 | 0.4138±0.0011 | 0.3931±0.0015 | 0.5731±0.0021 | 0.7828±0.0013 | 0.2136±0.0008 |
| TIMEX | 0.8324±0.0034 | 0.7219±0.0031 | 0.6381±0.0022 | 0.7124±0.0017 | 0.9411±0.0006 | 0.3380±0.0014 |
| TIMEX++ | 0.8905±0.0018 | 0.7805±0.0014 | 0.6618±0.0019 | 0.8468±0.0014 | 0.9069±0.0003 | 0.4064±0.0011 |
| ORTE | **0.9998**±0.0001 | **0.8269**±0.0014 | **0.8298**±0.0020 | **0.9001**±0.0025 | **0.9711**±0.0006 | **0.4503**±0.0031 |

| METHOD | SEQCOMB-MV | | | LOWVAR | | |
|---|---|---|---|---|---|---|
| | AUPRC | AUP | AUR | AUPRC | AUP | AUR |
| IG | 0.3298±0.0015 | 0.7483±0.0027 | 0.2581±0.0028 | 0.8691±0.0035 | 0.4827±0.0029 | 0.8165±0.0016 |
| DYNAMASK | 0.3136±0.0019 | 0.5481±0.0053 | 0.1953±0.0025 | 0.1391±0.0012 | 0.1640±0.0028 | 0.2106±0.0018 |
| WINIT | 0.2809±0.0018 | 0.7594±0.0024 | 0.2077±0.0021 | 0.1667±0.0015 | 0.1140±0.0022 | 0.3842±0.0017 |
| CORTX | 0.3629±0.0021 | 0.5625±0.0006 | 0.3457±0.0017 | 0.4983±0.0014 | 0.3281±0.0027 | 0.4711±0.0013 |
| SGT + GRAD | 0.4893±0.0005 | 0.4970±0.0005 | 0.4289±0.0018 | 0.3449±0.0010 | 0.2133±0.0029 | 0.3528±0.0015 |
| TIMEX | 0.6878±0.0021 | 0.8326±0.0008 | 0.3872±0.0015 | 0.8673±0.0033 | 0.5451±0.0028 | 0.9004±0.0024 |
| TIMEX++ | 0.7589±0.0014 | 0.8783±0.0007 | 0.3906±0.0011 | 0.9466±0.0015 | **0.8057**±0.0016 | 0.8332±0.0016 |
| ORTE | **0.8314**±0.0019 | **0.9011**±0.0005 | **0.5632**±0.0028 | **0.9637**±0.0016 | 0.7390±0.0023 | **0.9057**±0.0013 |

*Table 2.* (Left) Attribution explanation performance on the ECG dataset. (Right) Results of ablation analysis.

| METHOD | ECG | | | ORTE ABLATIONS | ECG | | |
|---|---|---|---|---|---|---|---|
| | AUPRC | AUP | AUR | | AUPRC | AUP | AUR |
| IG | 0.4182±0.0014 | 0.5949±0.0023 | 0.3204±0.0012 | FULL | **0.7183**±0.0019 | **0.8222**±0.0021 | 0.5133±0.0016 |
| DYNAMASK | 0.3280±0.0011 | 0.5249±0.0030 | 0.1082±0.0080 | replace STE | 0.6152±0.0007 | 0.7468±0.0008 | 0.4023±0.0012 |
| WINIT | 0.3049±0.0011 | 0.4431±0.0026 | 0.3474±0.0011 | w/o $\mathcal{L}_{JS}$ | 0.6946±0.0040 | 0.7941±0.0045 | 0.5208±0.0036 |
| CORTX | 0.3735±0.0008 | 0.4968±0.0024 | 0.3031±0.0008 | w/o $\mathcal{L}_{ct}$ | 0.5421±0.0032 | 0.7195±0.0028 | 0.4143±0.0019 |
| SGT + GRAD | 0.3144±0.0010 | 0.4241±0.0024 | 0.2639±0.0013 | w/o $\mathcal{L}_{con}$ | 0.7130±0.0037 | 0.8019±0.0037 | 0.4244±0.0021 |
| TIMEX | 0.4721±0.0018 | 0.5663±0.0025 | 0.4457±0.0018 | w/o $\mathcal{L}_{sim}$ | 0.7017±0.0040 | 0.7951±0.0043 | 0.5187±0.0035 |
| TIMEX++ | 0.6599±0.0009 | 0.7260±0.0010 | 0.4595±0.0007 | w/o $\mathcal{L}_{spr}$ | 0.6907±0.0051 | 0.3507±0.0037 | **0.8185**±0.0044 |
| ORTE | **0.7183**±0.0019 | **0.8222**±0.0021 | **0.5133**±0.0016 | | | | |

(Liu et al., 2024) is an improved version of TIMEX based on the information bottleneck principle.

### 4.2. Performance on Synthetic and Real-world Datasets

**Synthetic datasets.** As shown in Table 1, we compare ORTE with existing competitive interpreters on both univariate and multivariate datasets. ORTE achieves the best performance in 10/12 cases (3 metrics across 4 datasets). Compared to the strongest baseline TIMEX++, ORTE improves AUPRC by an average of 7.48%, AUP by 1.84%, and AUR by 22.27%. Compared to the second strongest baseline TIMEX, ORTE improves AUPRC by an average of 19.61%, AUP by 15.38%, and AUR by 22.33%. Previous methods prioritized the precision of temporal pattern localization but overlooked the completeness of explanations, resulting in higher AUP but lower AUR. In contrast, ORTE

achieves an optimal trade-off between low redundancy and high completeness. When considering the global metric AUPRC, ORTE significantly improves ground-truth explanations over TIMEX++, with improvements of 12.27% on FreqShapes, 6.29% on SeqComb-UV, 9.55% on SeqComb-MV, and 1.81% on LowVar. This aligns with our emphasized principle of optimal information retention. We also provide experimental results on CNN and LSTM in Appendix E and visualizations in Appendix H.

**Occlusion experiments on real-world datasets.** To evaluate the explanations of ORTE on real-world datasets lacking ground-truth, we follow the approach of TIMEX (Queen et al., 2024) and TIMEX++ (Liu et al., 2024) by masking the bottom $p$-percentile of the salient features and measuring the change in prediction AUROC. As shown in Figure 3, ORTE outperforms other methods on the PAM and Boiler

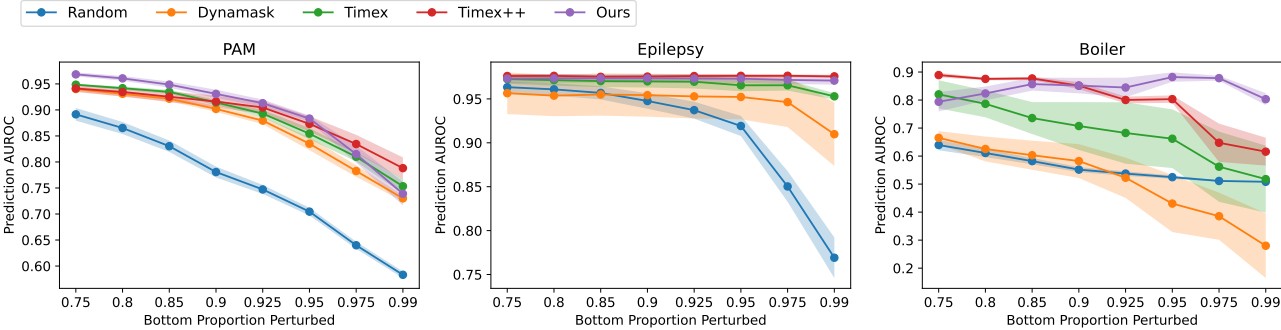

*Figure 3.* Occlusion experiments on real-world datasets. Higher values indicate better performance.

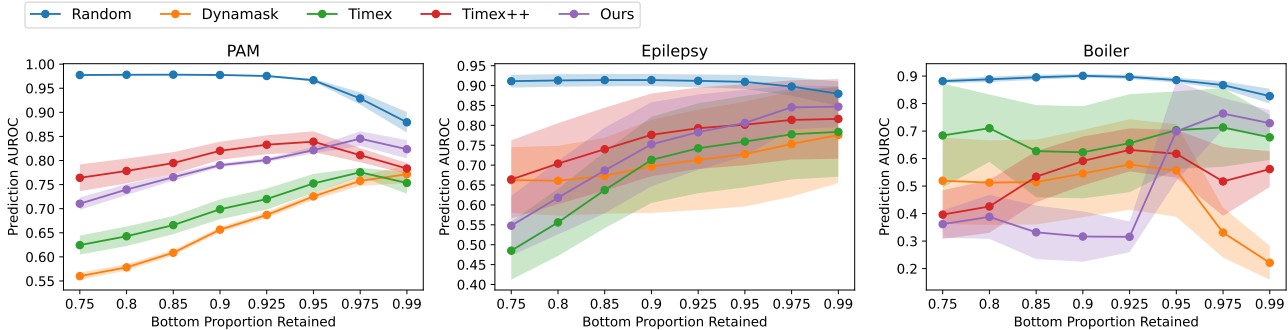

*Figure 4.* Insertion experiments on real-world datasets. Higher values indicate better performance.

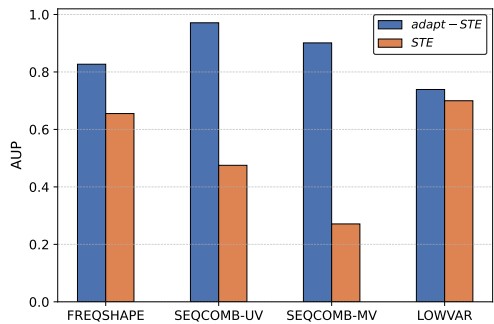

*Figure 5.* Comparison performance of AUP between *adapt*-STE and STE.

datasets, and performs comparably to the strongest baseline TIMEX++ on the Epilepsy dataset, both maintaining a high and stable level. Notably, ORTE demonstrates exceptional stability on the Boiler dataset, retaining a high AUROC even after masking 97.5% of the bottom features. Furthermore, compared to other baseline methods, ORTE exhibits narrower error bars, indicating that the ORTE method, guided by the principle of optimal information retention, also achieves outstanding stability.

**Insertion experiments on real-world datasets.** As a complement to the occlusion experiments, the insertion experiment inserts features from bottom to top, as shown in Figure

4. The results show that ORTE achieves a lower AUPROC when inserting the bottom 75% of the salient features. As the insertion percentage increases, the predicted AUROC gradually improves. When the insertion ratio reaches 97.5%, ORTE attains the highest AUPROC. This indicates that the most interpretable or informative features are concentrated in the high-saliency features, further validating the claims of low redundancy and high completeness. In addition, the Random baseline always maintains a high AUROC, which is because the random selection of features does not distinguish the salient or importance of features and contains informative time points or segments.

### 4.3. Ablation and Contrast Experiments

**Results and Ablation Experiments on ECG Data.** To directly demonstrate ORTE's performance on real-world data, we compared it with baseline methods in the application of arrhythmia detection using electrocardiogram (ECG) data. As shown in Table 2, the attribution maps of ORTE exhibit state-of-the-art performance in identifying the relevant QRS intervals driving arrhythmia diagnosis, outperforming the strongest baseline TIMEX++ by 8.85% (AUPRC), 13.25% (AUP), and 11.71% (AUR). Notably, ORTE achieves an overall improvement in both AUP and AUR, ensuring the identification of larger segments of the QRS interval rather than individual time steps.

To validate the contribution of each component in the ORTE method, we conducted ablation experiments, as shown on the right side of Table 2, where "w/o" indicates the absence of the relevant component. The results demonstrate that *adapt*-STE improves AUPRC by 7.27% and AUR by 14.22%, proving the effectiveness of *adapt*-STE. We also compared the performance of every term of the objective function. Compared to the full model, the absence of other loss functions leads to inferior explanations. The proposed ORTE effectively compensates for the limitations of individual components, achieving an overall optimal explanation.

**Contrast Experiments of *adapt*-STE.** We compare the performance of *adapt*-STE with STE on univariate and multivariate datasets, as shown in Figure 5. The AUP values of the four datasets show that *adapt*-STE is significantly better than STE. Moreover, AUP is considered to be the more important evaluation metric. Especially in SeqComb-UV and SeqComb-MV, *adapt*-STE can detect the key time patterns more accurately, which also verifies the better adaptation of *adapt*-STE to data diversity. The complete experiment results are provided in Appendix G.

## 5. Conclusion

This paper proposes an information-theoretic principle of optimal information retention to guide the discovery of explanatory temporal patterns in time series. It defines semantic information, redundant information, and effective information, and derives corresponding objective functions. Furthermore, we introduce ORTE as a practical implementation of this principle. Experiments on synthetic and real-world datasets demonstrate that the proposed method significantly outperforms competitive baseline approaches. We also believe that the optimal information retention principle proposed in this paper is applicable to the optimization and improvement of future interpretability methods.

**Limitations.** This study has two main limitations. First, the control of hyperparameters can affect instance-level explanations, requiring trial and adjustment based on the type and characteristics of the data. Second, the practical application of the optimal information retention principle to guide other interpretability methods requires further research. Extending it to gradient-based or mask-based explanation methods could be an interesting direction.

## Acknowledgements

This work was supported by the National Natural Science Foundation of China (Grant No. 62372031).

## Impact Statement

This paper presents work whose goal is to advance the field of Machine Learning. There are many potential societal consequences of our work, none which we feel must be specifically highlighted here.

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

# A. Related Work

**Deep Model Interpretability.** Current efforts to enhance the interpretability of deep models primarily focus on exploring the relationship between inputs and predictions or making the prediction process more transparent (Räuker et al., 2023). Specifically, these research endeavors can be categorized into four primary directions: i) Employing visualization techniques to reveal the response relationship between feature maps at any layer of the model and the input (Yosinski et al., 2015; Zeiler & Fergus, 2014). ii) Extracting rules or feature interactions, such as leveraging attention mechanisms to understand the role of each variable in predictions and identifying the main features driving the model (Choi et al., 2016; Qin et al., 2017). iii) Simulating the predictive behavior of deep models with interpretable models, such as approximating the decision surface of the model in the neighborhood of samples (Bhatt et al., 2020; Ribeiro et al., 2016; Guidotti et al., 2019). iv) Scoring the importance of input data and visualizing attribution maps (saliency maps or importance maps), which is currently the most active area of interpretability research, with representative methods like Integrated Gradients (IG) (Baehrens et al., 2010), Grad-CAM (Selvaraju et al., 2017), and SHAP (Lundberg, 2017). v) Mask-based methods are devoted to learning a mask matrix $M$, and the Hadamard products of the input and $M$ are represented as the salient features (Dabkowski & Gal, 2017; Chen et al., 2018; Fong et al., 2019). Jethani et al. point out that explanation methods suffer from computational efficiency, inaccuracy, or lack of faithfulness. The lack of robustness of the underlying black-box models, especially to the erasure of unimportant distractor features in the input is a key reason why certain attributions lack faithfulness (Bhalla et al., 2023). ROAR (Hooker et al., 2019) points out that many popular interpretability methods produce feature importance estimates that are no better than randomly assigned feature importance, and proposes an empirical measure of the approximate accuracy of feature importance estimates in deep neural networks. Although various interpretation methods have been applied to Computer Vision (CV) and Natural Language Processing (NLP), their application to time series often overlooks the modal heterogeneity. For instance, redundancy or incompleteness in time series interpretation may represent different temporal patterns.

**Time Series Interpretability.** Discriminative information in time series is often distributed across combinations of variables or multiple time steps. Designing specialized interpretation methods based on these characteristics of time series signals is the focus of current efforts (Doddaiah et al., 2022; Bento et al., 2021; Guidotti et al., 2020; Mujkanovic et al., 2020). A representative approach employs attention maps as references for interpreting deep models of time series. For example, WinIT (Leung et al., 2021) employs attention maps as references for feature importance, while Dynamask (Crabbé & Van Der Schaar, 2021) applies regularization constraints to attention maps to achieve smooth temporal patterns. However, these methods use zero-padding as a baseline for perturbation, which may generate out-of-distribution temporal patterns (Tonekaboni et al., 2020). The recent application of multiple instance learning to time series tasks has provided new insights for interpretability, i.e., Early et al. propose multiple instance learning for locally explainable time series classification, which guarantees the intrinsic interpretability of the deep model without compromising the predictive performance. Chen et al. have explored the feasibility of introducing multiple instance learning into time series tasks from the perspective of information theory and weakly supervised learning. Another category of methods involves using surrogate models to interpret pre-trained black-box models, such as TIMEX (Queen et al., 2024) and TIMEX++ (Liu et al., 2024). However, TIMEX is limited by the out-of-distribution shifts of the surrogate model, while TIMEX++ overlooks the completeness of temporal patterns, thus failing to capture the full temporal dynamics.

**Explanations of Information Theory.** Information theory provides a novel perspective for explaining the complex dynamics of deep models. Representative methods include information bottleneck (Tishby & Zaslavsky, 2015), contrastive learning (Dosovitskiy et al., 2014; Khosla et al., 2020; Tian et al., 2020), and rate reduction (Yu et al., 2020; Chan et al., 2022). Among these, the information bottleneck is used to interpret the learning process of deep models, describing it structurally as a trade-off between compact representation and information utility. Contrastive learning formalizes the pre-training objective as minimizing the distance between positive pairs in the embedding space while maximizing the distance between negative pairs, such as maximizing mutual information for positive samples and minimizing it for negative samples. Rate reduction posits that deep models are stacks of information compression and discretization (Yu et al., 2023). Although these methods offer insights into the foundational dynamics of deep models, they lack intuitive interpretability at the data level, meaning they cannot directly guide the optimization of local explanations.

# B. Technical Details from Section 2

We wish to reiterate the core contributions of our method from a technical perspective. Compared to heuristic interpretability methods, we first propose an optimal information retention principle from an information theory perspective and then

construct a practical framework based on the principle. The principle provides optimization directions for explanations across three levels: **fidelity**, **low redundancy**, and **high completeness**. The specific contributions of the principle are three-fold:

- We align the prediction distributions of the original time series and explanations within the pre-trained black-box model to ensure the fidelity of the explanation.

- We employ variational methods to learn a mask matrix that follows Bernoulli distributions, masking task-irrelevant information to ensure low redundancy in the explanation.

- We deconstruct the time series signal into positive and negative samples pairs, employing contrastive learning to ensure that positive samples contain task-relevant effective information while negative samples contain only irrelevant information, thereby ensuring the completeness of the explanation.

The second and third terms can naturally derive objective functions from the explicit conditional mutual information in the optimal information retention principle, and we provide the corresponding proofs below.

### B.1. Proof of Theorem 1

**Minimum Redundant Information Retention.** The optimal sub-features $X^m$ should only contain the mutual information shared by $I(X;Y)$, and include as little information of $X$ no relation to $Y$, as follows:

$$\min I\left(X; X^m \mid Y\right). \tag{16}$$

We aim to derive from $I\left(X; X^m \mid Y\right)$ the objective function for optimizing time-series explanations:

$$\mathcal{L}_{\text{mask}} = \mathbb{E}[D_{\text{KL}}(p(m \mid x)\|q(m))]$$
$$= \sum_{t,d}[p_{t,d} \log \frac{p_{t,d}}{r} + (1 - p_{t,d}) \log \frac{1 - p_{t,d}}{1 - r}]. \tag{17}$$

*Proof.*

$$I\left(X; X^m \mid Y\right) = \int_{\mathcal{Y}} \left( \int_{\mathcal{X}^{\mathcal{M}}} \int_{\mathcal{X}} \log \left( \frac{p\left(x, x^m \mid y\right)}{p(x \mid y)p\left(x^m \mid y\right)} \right) p\left(x, x^m \mid y\right) dx dx^m \right) p(y)dy$$
$$= \int_{\mathcal{Y}} \left( \int_{\mathcal{X}^{\mathcal{M}}} \int_{\mathcal{X}} \log \left( \frac{p\left(x^m \mid x, y\right)}{p\left(x^m \mid y\right)} \right) p\left(x, x^m \mid y\right) dx dx^m \right) p(y)dy$$
$$= \int_{\mathcal{Y}} \int_{\mathcal{X}^{\mathcal{M}}} \int_{\mathcal{X}} \log \left( \frac{p\left(x^m \mid x, y\right)}{p\left(x^m \mid y\right)} \right) p\left(x, x^m, y\right) dx dx^m dy \tag{18}$$
$$= \mathbb{E}_{p(x,x^m,y)} \left[ \log \frac{p\left(x^m \mid x, y\right)}{p\left(x^m \mid y\right)} \right]$$
$$= \mathbb{E}_{p(x,y)} \left[ \mathbb{E}_{p(x^m|x,y)} \left[ \log \frac{p\left(x^m \mid x, y\right)}{p\left(x^m \mid y\right)} \right] \right],$$

and we can view $\mathbb{E}_{p(x^m|x,y)} \left[ \log \frac{p(x^m|x,y)}{p(x^m|y)} \right]$ as the Kullback-Leibler divergence of $p(x^m \mid x, y)$ and $p(x^m \mid y)$. So $I\left(X; X^M \mid Y\right)$ can be rewritten as:

$$I\left(X; X^M \mid Y\right) = \mathbb{E}_{p(x,y)} \left[ D_{\text{KL}}(p(x^m \mid x, y)\|p(x^m \mid y)) \right] \tag{19}$$

where $x^m = x \odot m$ and (19) can be redefined as:

$$I\left(X; M \mid Y\right) = \mathbb{E}_{p(x,y)} \left[ D_{\text{KL}}(p(m \mid x, y)\|p(m \mid y)) \right]. \tag{20}$$

So there are

$$\min I\left(X; X^M \mid Y\right) \propto \min D_{\text{KL}}(p(m \mid x, y)\|p(m \mid y)). \tag{21}$$

Since $p(m \mid y)$ is intractable, we introduce a variational approximation $q(m \mid y)$ for this problem. We now proceed in two steps. First, we deal with the $D_{KL}$ minimization problem without $y$ as a conditional constraint. Considering that any point $m_{i,j} \in \mathbb{R}^{T \times D}$ on $M$ follows a Bernoulli distribution with value $v = \{0, 1\}$, so the distribution of $M$ can be assumed as:

$$q(m) = \prod_{t,d} \text{Bern}(r), \tag{22}$$

where $r$ as the desired probability of $v = 1$ is a hyperparameter in practice. Similarly, we parameterize the learnable multiple Bernoulli distribution of $p(m \mid x)$ as:

$$p(m \mid x) = \prod_{t,d} \text{Bern}(p_{t,d}), \tag{23}$$

where $p_{t,d}$ denotes the learned probability of $x_{t,d}$ is preserved.

Second, given the condition constraint $y$, the expected distribution $q(m \mid y)$ or $p(m \mid x, y)$ should implement the conditional probability with $y$ as a known term. We approximate the conditional constraint by considering the distributional consistency of $(y, \hat{y})$ as a complement to the conditional probability, as follows:

$$\mathcal{L}_{\text{JS}}(f(x), f(x^m)) = \mathbb{E}[D_{\text{JS}}(f(x) \| f(x^m))], \tag{24}$$

where $D_{\text{JS}}$ represents the Jensen-Shannon (JS) divergence. And $\mathcal{L}_{\text{JS}}(f(x), f(x^m)) \approx 0$ is the ideal case. Combined with the analysis in the first step, the objective function derived from $\min I(X; M \mid Y)$ can be described as:

$$\min \mathbb{E}[D_{\text{KL}}(p(m \mid x) \| q(m))], \text{ s.t. }, \mathcal{L}_{\text{LC}}(f(x), f(x^m)) \approx 0. \tag{25}$$

Since $\mathcal{L}_{\text{JS}}(f(x), f(x^m)) \geq 0$, so the constraint term combined in one function:

$$\min\{\mathbb{E}[D_{\text{KL}}(p(m \mid x) \| q(m))] + \mathcal{L}_{\text{LC}}(f(x), f(x^m))\}, \tag{26}$$

where

$$\mathbb{E}[D_{\text{KL}}(p(m \mid x) \| q(m))] = \sum_{t,d}[p_{t,d} \log \frac{p_{t,d}}{r} + (1 - p_{t,d}) \log \frac{1 - p_{t,d}}{1 - r}]. \tag{27}$$

Furthermore, considering that the second term is the objective function corresponding to *Semantic Information Retention*, (26) can be further simplified to the minimization of the objective function:

$$\begin{aligned} \mathcal{L}_{\text{mask}} &= \mathbb{E}[D_{\text{KL}}(p(m \mid x) \| q(m))] \\ &= \sum_{t,d}[p_{t,d} \log \frac{p_{t,d}}{r} + (1 - p_{t,d}) \log \frac{1 - p_{t,d}}{1 - r}]. \end{aligned} \tag{28}$$

**Summary.** We have shown the objective function (17) derived from $\min I(X; X^M \mid Y)$. This completes the proof of definition. $\square$

### B.2. Proof of Theorem 2

**Maximum Effective Information Retention.** The optimal retention of sub-features $X^m$ should encompass as much task-relevant information from $X$ as possible, minimizing the risk of information omission, as follows:

$$\min I(X; Y \mid X^m). \tag{29}$$

We aim to derive from $I(X; Y \mid X^m)$ the objective function for optimizing time-series explanations:

$$L_{\text{cont}} = -\frac{1}{N} \sum_{i=1}^{N} \log \frac{\exp(f(x_i) \cdot f(\hat{x}_i))}{\exp(f(x_i) \cdot f(\hat{x}_i)) + \exp(f(x_i) \cdot f(\hat{x}_j^-))}. \tag{30}$$

*Proof.*

$$
\begin{aligned}
I\left(X;Y\mid X^{m}\right) &= \int_{\mathcal{X}^m}\left(\int_y\int_{\mathcal{X}}\log\left(\frac{p\left(x,y\mid x^m\right)}{p\left(x\mid x^m\right)p\left(y\mid x^m\right)}\right)p\left(x,\mathrm{y}\mid x^m\right)dxdy\right)p\left(x^m\right)dx^m\\
&= \int_{\mathcal{X}^m}\int_y\int_{\mathcal{X}}\log\left(\frac{p\left(x,y,x^m\right)p(x)p\left(x^m\right)}{p(x)p\left(y,x^m\right)p\left(x,x^m\right)}\right)p\left(x,\mathrm{y},x^m\right)dxdydx^m\\
&= \int_{\mathcal{X}^m}\int_y\int_{\mathcal{X}}\left[\log\frac{p\left(x,y,x^m\right)}{p(x)p\left(y,x^m\right)}-\log\frac{p\left(x,x^m\right)}{p(x)p\left(x^m\right)}\right]p\left(x,\mathrm{y},x^m\right)dxdydx^m\\
&= \int_{\mathcal{X}^m}\int_y\int_{\mathcal{X}}\log\frac{p\left(x,y,x^m\right)}{p(x)p\left(y,x^m\right)}p\left(x,\mathrm{y},x^m\right)dxdydx^m - \int_{\mathcal{X}^m}\int_{\mathcal{X}}\log\frac{p\left(x,x^m\right)}{p(x)p\left(x^m\right)}p\left(x,x^m\right)dxdx^m\\
&= I\left(X;Y,X^M\right)-I\left(X;X^M\right),
\end{aligned}
\tag{31}
$$

and since $X^m$, as a sub-segment of $X$, cannot be directly used as input to $f(\cdot)$ for the different shapes. Therefore, $\hat{X} = g(X^m)$ is used as new input to quantify the conditional mutual information $I\left(X;Y\mid X^m\right)$, assuming $distribution(\hat{X})\sim distribution(X)$. $g$ is the generator function, i.e., one layer of MLP. Then, there are

$$
I\left(X;Y\mid X^{m}\right)\Rightarrow I\left(X;Y\mid\hat{X}\right)=I\left(X;Y,\hat{X}\right)-I\left(X;\hat{X}\right).
\tag{32}
$$

To minimize $I\left(X;Y\mid X^m\right)$, we minimize the first term $I\left(X;Y,\hat{X}\right)$, while maximize the second term $I(X;\hat{X})$. When $\hat{X}$ serves as the input generated by $X^m$, the mutual information $I(X;\hat{X})$ should ideally include only the information contained in $I(X;Y)$. Otherwise, redundant information may be introduced. Assuming $\hat{X}$ has no overlap with the portions of $H(X\mid Y)$, then $I\left(X;Y,\hat{X}\right)=I\left(X;Y\right)$, which is a constant independent of $\hat{X}$.

For $I(X;\hat{X})$, since $X$ and $\hat{X}$ share the effective information in $X^m$, they can naturally regarded as the anchor sample and the positive sample, respectively, in contrastive learning. Therefore, we employ the contrastive loss including three to maximize the mutual information between $X$ and $\hat{X}$:

$$
L_{\mathrm{cont}} = -\frac{1}{N}\sum_{i=1}^{N}\log\frac{\exp\left(f\left(x_i\right)\cdot f\left(\hat{x}_i\right)\right)}{\exp\left(f\left(x_i\right)\cdot f\left(\hat{x}_i\right)\right)+\exp\left(f\left(x_i\right)\cdot f\left(\hat{x}_j^-\right)\right)},
\tag{33}
$$

where $\hat{x}_j^-$ denotes the negative sample, which can be generated by $h(X-X^m)$, i.e., Gaussian noise imputation.

**Summary.** We have shown the objective function (30) derived from $\min I\left(X;Y\mid X^m\right)$. This completes the proof of definition. $\qquad\square$

## C. Pseudocode of *adapt*-STE

---
**Algorithm 1** *adapt*-STE Estimator

---
1: **Input:** Logits $l$, Forward temperature $\tau^f$, Backward temperature $\tau^b$, Updating rate $\eta$, Standard deviation $\sigma$, Upper bound of temperature $\tau_{max}$, Lower bound of temperature $\tau_{min}$
2: **Forward Pass:**
3:    Sample Gumbel noise: $g\sim\mathrm{Gumbel}(0,1)$
4:    Update forward temperature: $\tau_f=\mathrm{clip}\left(\tau_f-\eta\cdot\sigma(l),\tau_{\min},\tau_{\max}\right)$
5:    Compute relaxed logits for forward pass: $p=\mathrm{softmax}\left(\frac{l}{\tau^f}+g\right)$
6:    Compute discrete mask: $m=\arg\max(p)$
7: **Backward Pass:**
8:    Update backward temperature: $\tau_b=\mathrm{clip}\left(\tau_b+\eta\cdot\sigma(l),\tau_{\min},\tau_{\max}\right)$
9:    Compute relaxed logits for backward pass using same Gumbel noise sample: $p^b=\mathrm{softmax}\left(\frac{l}{\tau^b}+g\right)$
10:    Compute gradient using $p^b$ instead of $p$
11:    Perform backpropagation using the relaxed gradients

---

# D. Exprimental Details

## D.1. Description of Datasets

We evaluate our method on four carefully designed synthetic datasets with ground-truth annotations and four real-world datasets. The number of samples, length, dimension, number of categories and tasks for all datasets are summarized in Table 3.

**Synthetic Datasets.** We employ four synthetic datasets with known ground-truth explanations proposed by Queen et al. (2024) to investigate the ability to identify interpretable time patterns.

- **FreqShapes** is a dataset designed for discerning the frequency of anomalous signal occurrences. It comprises four classes: Class 0 presents a downward pulse every 10 steps, Category 1 presents an upward pulse every 10 steps, Class 2 presents a downward pulse every 17 steps, and Class 3 presents an upward pulse every 17 steps. The ground truth is explained by the positions of the ascending and descending peaks.

- **SeqComb-UV** is a dataset tailored for distinguishing signal trends. It consists of four classes: Class 0 has no trend, Class 1 includes subsequences with an upward trend, Class 2 contains subsequences with a downward trend, and Class 3 simultaneously encompasses both upward and downward subsequences. The ground truth is explained by identifying the locations of the subsequences that exhibit trends.

- **SeqComb-MV** represents the multivariate version of SeqComb-UV. Subsequences with trends are randomly injected into the sensors. The ground truth is also explained by pinpointing the positions of the subsequences that exhibit trends.

- **LowVar** is a dataset aimed at identifying regions of low variance that change over time within multivariate time series samples. It is divided into four classes: Class 0 identifies discriminant subsequences with a mean of -1.5 on sensor 0; Class 1 identifies discriminant subsequences with a mean of 1.5 on sensor 0; Class 2 identifies discriminant subsequences with a mean of -1.5 on sensor 1; and Class 3 identifies discriminant subsequences with a mean of 1.5 on sensor 1. The ground truth is explained by determining the locations of the discriminant subsequences.

**Real-world Datasets.** We employed four real-world datasets, among which only the ECG dataset has generated ground-truth explanations.

- **ECG** (Moody & Mark, 2001) is a dataset used for classifying cardiac arrhythmias. It includes three classes: normal readings (N), left bundle branch block beats (L), and right bundle branch block beats (R). The original dataset was segmented into 92,511 samples by windowing, each containing 360 time steps. We selected these because it is known that the L and R diagnoses depend on the QRS interval, which will serve as our ground-truth interpretation.

- **PAM** (Reiss & Stricker, 2012) is a dataset utilized for discerning human activities. It encompasses 8 classes, each representing a different daily life activity. Every sample comprises 17 sensors with 600 consecutive observations, sampled at a frequency of 100 Hz, and the samples are roughly balanced across all 8 classes.

- **Epilepsy** (Andrzejak et al., 2001) is an electroencephalogram (EEG) dataset designed to distinguish epileptic seizures. It contains two categories: seizure and non-seizure. Each sample lasts for 1 second and is sampled at a frequency of 178 Hz.

- **Boiler** (Shohet et al., 2019) is a dataset from an industrial setting. The learning task involves detecting mechanical failures in the blowdown valves of boilers. This dataset is particularly challenging due to its large dimensionality ratio, unlike other datasets that contain more time steps than sensors.

## D.2. Description of Baselines

- **IG.** (Baehrens et al., 2010) Integrated Gradients (IG) utilizes the gradients of a model to derive attribution maps for explanation. This method compares gradients to a baseline value and performs Riemann integration. Although Integrated Gradients is a classical explanation method, it does not incorporate inductive biases specific to time series.

- **SGT + Grad.** (Ismail et al., 2021) This method employs Saliency Guided Training. During the training process, features with low gradients are masked to guide the model to focus on more significant regions for prediction.

Table 3. The description of synthetic and real-world datasets.

| DATASET | # OF SAMPLES | LENGTH | DIMENSION | CLASSES | TASK |
|---------|-------------|--------|-----------|---------|------|
| FREQSHAPES | 6,100 | 50 | 1 | 4 | MULTI-CLASSIFICATION |
| SEQCOMB-UV | 6,100 | 200 | 1 | 4 | MULTI-CLASSIFICATION |
| SEQCOMB-MV | 6,100 | 200 | 4 | 4 | MULTI-CLASSIFICATION |
| LOWVAR | 6,100 | 200 | 2 | 4 | MULTI-CLASSIFICATION |
| ECG | 92,511 | 360 | 1 | 5 | ECG CLASSIFICATION |
| PAM | 5,333 | 600 | 17 | 8 | ACTION RECOGNITION |
| EPILEPSY | 11,500 | 178 | 1 | 2 | EEG CLASSIFICATION |
| BOILER | 160,719 | 36 | 20 | 2 | MECHANICAL FAULT DETECTION |

- **Dynamask. (Crabbé & Van Der Schaar, 2021)** This explainer is specifically constructed for time series and generates explanations using perturbation-based iterations. The method iteratively occludes each sample to learn a continuous mask. The goal is to morph the input time series into a carefully determined baseline value, thereby highlighting important temporal patterns.

- **WinIT. (Leung et al., 2021)** Similar to Dynamask, this method is designed for time series and learns a mask matrix to achieve explanations. WinIT removes segments of the time series to measure the impact of features on the final prediction value. It considers feature dependencies across time steps and uses a generative model to alleviate Out-of-Distribution (OOD) issues.

- **CoRTX. (Chuang et al., 2023)** This method is an explainer that leverages contrastive learning to approximate SHAP values. Originally developed for computer vision, we adopted a custom version implemented in A, aiming for the time series encoder and explanation generator to work in concert.

- **TIMEX. (Queen et al., 2024)** This method trains an interpretable surrogate to mimic the behavior of a pre-trained time series model. It addresses the model fidelity issue by introducing model behavior consistency. It preserves the relationships in the latent space induced by the pre-trained model and those induced by TIMEX, learning a discrete mask matrix to explain time series.

- **TIMEX++. (Liu et al., 2024)** This method investigates from an information-theoretic perspective how existing interpretability methods may be affected by trivial solutions and distribution shift problems. To tackle these issues, TIMEX++ employs the information bottleneck principle to revise the objective function of TIMEX. This method is currently the most competitive baseline approach.

### D.3. Description of Metrics

As described by Crabbé & Van Der Schaar (2021), we employ AUP and AUR to evaluate the efficacy of explanation methods. As described above, let the binary mask matrix $M \in \{0, 1\}^{T \times D}$ be the explanation of any black-box models for time series, and let $Q \in \{0, 1\}^{T \times D}$ be the ground-truth mask matrix. If $Q_{t,d} = 1$, then the feature of $x_{t,d}$ is salient; Otherwise $Q_{t,d} = 0$, then the feature of $x_{t,d}$ is task-irrelevant. We aim to convert the mask into an estimator as follows:

$$\hat{Q}_{t,d}(\tau) = \begin{cases} 1 & \text{if } M_{t,d} \geq \tau \\ 0 & \text{else.} \end{cases} \tag{34}$$

Consider the saliency index obtained by the truth value and explanation method separately:

$$A = \{(t, d) \in [1 : T] \times [1 : D] \mid Q_{t,d} = 1\},$$
$$\hat{A}(\tau) = \{(t, d) \in [1 : T] \times [1 : D] \mid \hat{Q}_{t,d}(\tau) = 1\}. \tag{35}$$

The precision and recall curves that map each threshold to precision and recall scores are as follows:

$$P : (0, 1) \longrightarrow [0, 1] : \tau \longmapsto \frac{|A \cap \hat{A}(\tau)|}{|\hat{A}(\tau)|},$$

$$R : (0, 1) \longrightarrow [0, 1] : \tau \longmapsto \frac{|A \cap \hat{A}(\tau)|}{|A|}. \tag{36}$$

So AUP and AUR can be derived by:

$$AUP = \int_0^1 P(\tau)d\tau,$$

$$AUR = \int_0^1 R(\tau)d\tau. \tag{37}$$

## D.4. Description of Black-box Hyperparameters

In order to fairly compare the effectiveness of interpretation methods, it is necessary to ensure that the parameters of the pre-trained black-box models are consistent. To this end, we refer to the parameter settings of Liu et al. (2024) as shown in Table 4.

Table 4. Training parameters for transformer-based predictors across all ground-truth and real-world datasets.

| PARAMETER | FREQSHAPE | SEQCOMB-UV | SEQCOMB-MV | LOWVAR | ECG | PAM | EPILEPSY | BOILER | WAFER | FREEZERREGULAR | WATER |
|---|---|---|---|---|---|---|---|---|---|---|---|
| LEARNING RATE | 0.001 | 0.001 | 0.0005 | 0.001 | 0.002 | 0.001 | 0.0001 | 0.001 | 0.0001 | 0.0001 | 0.002 |
| WEIGHT DECAY | 0.1 | 0.01 | 0.001 | 0.01 | 0.001 | 0.001 | 0.001 | 0.001 | 0.001 | 0.001 | 0.001 |
| EPOCHS | 100 | 200 | 1000 | 120 | 500 | 100 | 300 | 500 | 200 | 300 | 500 |
| NUM. LAYERS | 1 | 2 | 2 | 1 | 1 | 1 | 1 | 1 | 1 | 1 | 1 |
| $d_h$ | 16 | 64 | 128 | 32 | 64 | 72 | 16 | 32 | 16 | 16 | 64 |
| DROPOUT | 0.1 | 0.25 | 0.25 | 0.25 | 0.1 | 0.25 | 0.1 | 0.25 | 0.1 | 0.1 | 0.1 |
| NORM. EMBEDDING | NO | NO | NO | YES | YES | NO | NO | YES | NO | NO | YES |

## D.5. Description of ORTE Hyperparameters

Table 5 summarizes the parameter settings for ORTE, where $\alpha$ is utilized to regulate the impact of contrastive learning on the interpretability effect. When $\alpha$ is optimized, it preserves more complete interpretable temporal patterns, avoiding the loss of valuable information. $\beta$ is employed to prevent overfitting and enhance generalization capabilities on test data. $\gamma$ ensures the sparsity of explanations while also serving as a supplement to low redundancy. $\eta$, $\tau_{max}$ and $\tau_{min}$ are the hyperparameters corresponding to *adapt*-STE, where we adopted the same parameter settings to ensure applicability to different datasets. All the experiments are performed on Ubuntu 18.04.6 LTS and 4 GPU NVIDIA GeForce RTX 2080.

Table 5. Training parameters for ORTE across all ground-truth and real-world experiments.

| PARAMETER | FREQSHAPE | SEQCOMB-UV | SEQCOMB-MV | LOWVAR | ECG | PAM | EPILEPSY | BOILER |
|---|---|---|---|---|---|---|---|---|
| LEARNING RATE | 0.001 | 0.001 | 0.002 | 0.005 | 0.0005 | 0.0005 | 0.0005 | 0.0001 |
| BATCH SIZE | 64 | 64 | 64 | 64 | 16 | 32 | 32 | 32 |
| WEIGHT DECAY | 0.001 | 0.001 | 0.001 | 0.0001 | 0.0001 | 0.001 | 0.001 | 0.001 |
| SCHEDULER | YES | YES | NO | NO | NO | NO | YES | YES |
| EPOCHS | 50 | 50 | 100 | 100 | 5 | 100 | 50 | 50 |
| $r$ | 0.5 | 0.5 | 0.5 | 0.5 | 0.5 | 0.1 | 0.5 | 0.5 |
| $\eta$ | 0.01 | 0.01 | 0.01 | 0.01 | 0.01 | 0.01 | 0.01 | 0.01 |
| $\tau_{max}$ | 3 | 3 | 3 | 3 | 3 | 3 | 3 | 3 |
| $\tau_{min}$ | 1 | 1 | 1 | 1 | 1 | 1 | 1 | 1 |
| $\alpha$ | 10.0 | 5.0 | 10.0 | 1.0 | 10.0 | 10.0 | 2.0 | 10.0 |
| $\beta$ | 1.0 | 1.0 | 1.0 | 2.0 | 1.0 | 1.0 | 1.0 | 1.0 |
| $\gamma$ | 0.05 | 0.05 | 0.05 | 0.05 | 0.05 | 0.01 | 0.005 | 0.001 |

## D.6. Description of classification Performance

Excellent classification performance is a necessary guarantee for evaluating the effectiveness of explanations. To objectively compare interpretation methods, we employ the vanilla Transformer (Vaswani, 2017) as the black-box classifier. Table 6 summarizes the classification performance of the Transformer on four synthetic datasets and four real-world datasets. The results show outstanding performance on all datasets except for dataset Boiler, with both AUP and AUR exceeding 0.9. The reason is that dataset Boiler is more challenging due to its larger dimension-to-length ratio, unlike the other datasets, which contain more time steps than sensors.

*Table 6.* The performance of transformer-based predictors for time series classification.

| DATASET | F1 | AUPRC | AUROC |
|---|---|---|---|
| FREQSHAPES | 0.9692±0.0055 | 0.9936±0.0026 | 0.9980±0.0008 |
| SEQCOMB-UV | 0.9412±0.0057 | 0.9789±0.0034 | 0.9923±0.0011 |
| SEQCOMB-MV | 0.9270±0.0548 | 0.9445±0.0510 | 0.9792±0.0194 |
| LOWVAR | 0.9854±0.0044 | 0.9978±0.0006 | 0.9992±0.0003 |
| ECG | 0.9211±0.0311 | 0.9463±0.0320 | 0.9709±0.0165 |
| PAM | 0.8876±0.0064 | 0.9312±0.0044 | 0.9785±0.0015 |
| EPILEPSY | 0.9260±0.0114 | 0.9383±0.0079 | 0.9630±0.0135 |
| BOILER | 0.8363±0.0178 | 0.8214±0.0263 | 0.8901±0.0348 |

# E. Additional Experiments on CNN and LSTM

We now study the applicability of ORTE across different black-box model architectures. Specifically, we conducted experiments on both convolutional neural networks (CNN) and long-short term memory (LSTM) networks, which are specifically structured as follows:

- CNN: 3 layer CNN + MLP + meanpool.

- LSTM: 3 layer bidirectional LSTM + MLP + mean of last hidden states.

The classification Performance of CNN and LSTM are summarized in Table 7 and Table 8, respectively.

*Table 7.* The performance of CNN-based predictors for time series classification.

| DATASET | F1 | AUPRC | AUROC |
|---|---|---|---|
| FREQSHAPES | 0.9910±0.0071 | 0.9997±0.0007 | 0.9999±0.0002 |
| SEQCOMB-MV | 0.9833±0.0171 | 0.9981±0.0024 | 0.9993±0.0009 |
| ECG | 0.9178±0.0119 | 0.9466±0.0136 | 0.9682±0.0052 |

*Table 8.* The performance of LSTM-based predictors for time series classification.

| DATASET | F1 | AUPRC | AUROC |
|---|---|---|---|
| FREQSHAPES | 0.9798±0.0141 | 0.9976±0.0018 | 0.9992±0.0007 |
| SEQCOMB-MV | 0.9332±0.1052 | 0.9473±0.1057 | 0.9824±0.0348 |
| ECG | 0.7907±0.1594 | 0.8095±0.1362 | 0.8303±0.1389 |

We compared ORTE with five competitive baseline methods: IG, Dynamask, WinIT, TIMEX, and TIMEX++. Table 9 summarizes the experimental performance of ORTE for explaining CNN models on both univariate and multivariate synthetic datasets. Table 10 summarizes the experimental performance of ORTE for explaining CNN models on ECG datasets as real-world applications. Overall, the interpretation effectiveness of ORTE shows significant improvement compared to competitive baseline methods. Specifically, relative to the currently most competitive interpretation method

TIMEX++, the AUP of ORTE increases by 36.86%, 0.31%, and 13.62% on datasets FreqShapes, Seqcomb-MV, and C, respectively. Furthermore, AUPRC respectively improved by 9.46%, 0.51%, and 26.23%, which aligns with the trade-off between low redundancy and high completeness advocated in this paper.

*Table 9.* Explainer results with CNN predictor on FREQSHAPES and SEQCOMB-MV synthetic datasets.

| METHOD | FREQSHAPES | | | SEQCOMB-MV | | |
|---|---|---|---|---|---|---|
| | AUPRC | AUP | AUR | AUPRC | AUP | AUR |
| IG | 0.9905±0.0007 | **0.8777**±0.0009 | 0.7056±0.0017 | 0.5979±0.0027 | 0.8858±0.0014 | 0.2294±0.0013 |
| DYNAMASK | 0.2574±0.0008 | 0.4432±0.0032 | 0.5257±0.0015 | 0.4550±0.0016 | 0.7308±0.0025 | 0.3135±0.0019 |
| WINIT | 0.5321±0.0018 | 0.6020±0.0025 | 0.3966±0.0017 | 0.5334±0.0011 | 0.8324±0.0020 | 0.2259±0.0020 |
| TIMEX | 0.7489±0.0046 | 0.4966±0.0033 | 0.7916±0.0021 | 0.7016±0.0019 | 0.7670±0.0012 | **0.4689**±0.0016 |
| TIMEX++ | 0.9134±0.0014 | 0.6066±0.0011 | 0.7952±0.0014 | 0.7822±0.0012 | 0.8896±0.0005 | 0.3434±0.0012 |
| ORTE | **0.9998**±0.0001 | 0.8302±0.0013 | **0.9997**±0.0001 | **0.7862**±0.0023 | **0.8924**±0.0005 | 0.3928±0.0020 |

*Table 10.* Explainer results with CNN predictor on ECG dataset.

| METHOD | ECG | | |
|---|---|---|---|
| | AUPRC | AUP | AUR |
| IG | 0.4949±0.0010 | 0.5374±0.0012 | **0.5306**±0.0010 |
| DYNAMASK | 0.4598±0.0010 | 0.7216±0.0027 | 0.1314±0.0008 |
| WINIT | 0.3963±0.0011 | 0.3292±0.0020 | 0.3518±0.0012 |
| TIMEX | 0.6401±0.0010 | 0.7458±0.0011 | 0.4161±0.0008 |
| TIMEX++ | 0.6726±0.0010 | 0.7570±0.0011 | 0.4319±0.0012 |
| ORTE | **0.8490**±0.0018 | **0.8601**±0.0016 | 0.4079±0.0012 |

Similarly, Table 11 summarizes the experimental performance of ORTE for explaining LSTM models on both univariate and multivariate synthetic datasets. Table 12 summarizes the experimental performance of ORTE for explaining LSTM models on ECG datasets as real-world applications. The experimental results for LSTM are also outstanding, especially on the multivariate dataset Seqcomb-MV, where AUPRC improved by 92%, AUP by 23.85%, and AUR by 27.25%, respectively.

*Table 11.* Explainer results with LSTM predictor on FREQSHAPES and SEQCOMB-MV synthetic datasets.

| METHOD | FREQSHAPES | | | SEQCOMB-MV | | |
|---|---|---|---|---|---|---|
| | AUPRC | AUP | AUR | AUPRC | AUP | AUR |
| IG | 0.9282±0.0016 | 0.7775±0.0010 | 0.6926±0.0017 | 0.2369±0.0020 | 0.5150±0.0048 | 0.3211±0.0032 |
| DYNAMASK | 0.2290±0.0012 | 0.3422±0.0037 | 0.5170±0.0013 | 0.2836±0.0021 | 0.6369±0.0047 | 0.1816±0.0015 |
| WINIT | 0.4171±0.0016 | 0.5106±0.0026 | 0.3909±0.0017 | 0.3515±0.0014 | 0.6547±0.0026 | 0.3423±0.0021 |
| TIMEX | 0.9903±0.0002 | 0.7887±0.0008 | 0.7963±0.0013 | 0.1298±0.0017 | 0.1307±0.0022 | **0.4751**±0.0015 |
| TIMEX++ | 0.9939±0.0002 | 0.7413±0.0009 | 0.8428±0.0008 | 0.4052±0.0038 | 0.6804±0.0052 | 0.3519±0.0021 |
| ORTE | **0.9999**±0.0000 | **0.8573**±0.0011 | **0.9993**±0.0003 | **0.7780**±0.0019 | **0.8427**±0.0004 | 0.4478±0.0015 |

*Table 12.* Explainer results with LSTM predictor on ECG dataset.

| METHOD | ECG | | |
|---|---|---|---|
| | AUPRC | AUP | AUR |
| IG | 0.5037±0.0018 | 0.6129±0.0026 | 0.4026±0.0015 |
| DYNAMASK | 0.3730±0.0012 | 0.6299±0.0030 | 0.1102±0.0007 |
| WINIT | 0.3628±0.0013 | 0.3805±0.0022 | 0.4055±0.0009 |
| TIMEX | 0.6057±0.0018 | 0.6416±0.0024 | 0.4436±0.0017 |
| TIMEX++ | 0.6512±0.0011 | 0.7432±0.0011 | 0.4451±0.0008 |
| ORTE | **0.7372**±0.0019 | **0.7817**±0.0020 | **0.5672**±0.0018 |

## F. Computation Cost

We evaluated the computation cost of interpretability methods on two real-world datasets (PAM and Epilepsy), as detailed in Table 13. Table 13 compares our method (ORTE) with four baseline approaches (IG, Dynamask, TIMEX, and TIMEX++) in terms of parameters (M), FLOPs (G), and Inference Runtime (s). Compared with TIMEX and Timex++, our inference time is slightly longer due to the double-STE to implement the regulation of discrete maps, but all three are in the same rank, i.e., less than 1 second. IG and Dynamask require longer inference time, which is consistent with expectations since both methods operate recursively on a single sample, while our method requires only one forward propagation.

*Table 13.* Comparison of computational costs.

| Method | PAM | | | Epilepsy | | |
|---|---|---|---|---|---|---|
| | Param(M) | FLOPS(G) | Inference runtime(s) | Param(M) | FLOPS(G) | Inference runtime(s) |
| IG | - | - | 8.7837 ± 0.1862 | - | - | 23.9806 ± 1.5245 |
| Dynamask | - | - | 458.4144 ± 32.5353 | - | - | 979.5168 ± 2.8082 |
| TimeX | 0.0620 | 6.5404 | 0.2475 ± 0.1072 | 0.0169 | 0.3546 | 0.2545 ± 0.1162 |
| TimeX++ | 2.7012 | 8.6530 | 0.2451 ± 0.1013 | 0.0601 | 0.4487 | 0.2586 ± 0.1176 |
| ORTE | 2.7097 | 6.7700 | 0.4914 ± 0.0053 | 0.0645 | 0.3585 | 0.5858 ± 0.0439 |

## G. Contrast Experiments of *adapt*-STE

To further validate the effectiveness of *adapt*-STE, we compared the performance of *adapt*-STE and STE on four datasets: FreqShapes, SeqComb-UV, SeqComb-MV, and LowVar. As shown in Figure 6, *adapt*-STE significantly outperforms STE in metrics of AUPRC, AUP, and AUR. Particularly in AUP, *adapt*-STE demonstrates higher explanation accuracy, which is considered a more critical metric. Compared to STE, *adapt*-STE exhibits a lower AUR on dataset FreqShapes, which may be attributed to an imbalance between low redundancy and high completeness, resulting in the omission of some effective information.

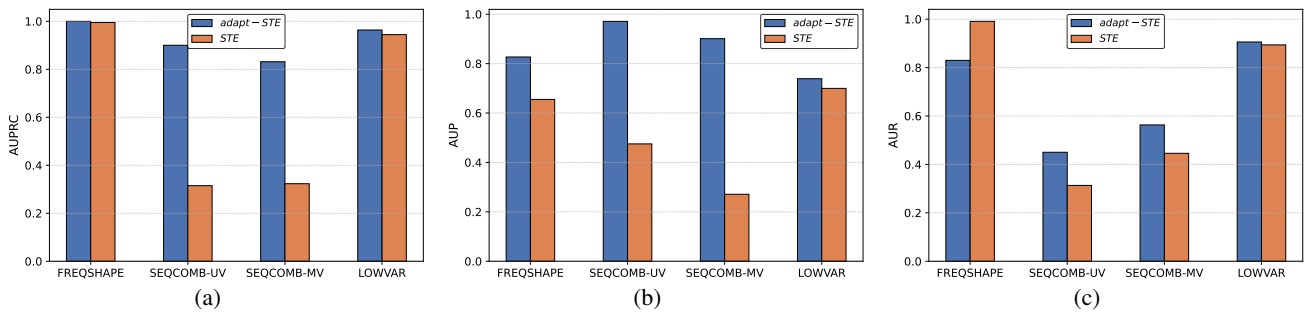

*Figure 6.* Comparison performance between *adapt*-STE and STE. (a) Comparison performance of AUPRC. (b) Comparison performance of AUP. (c) Comparison performance of AUR.

## H. Visual Comparison and Analysis

To facilitate a more intuitive comparison of various explanation methods, we have visualized the saliency maps of these methods. We randomly selected three samples from four datasets, each accompanied by ground truth explanations, to qualitatively analyze the effectiveness of the explanation methods through comparison with the ground truth. As shown in Figures 7-10, the results on datasets FreqShapes, SeqComb-UV, SeqComb-MV, and LowVar are visualized respectively. Each column represents a sample, and each row corresponds to an explanation method, with the bottom row representing the ground truth explanation. In the figures, darker colors indicate more significant features, with the ground truth explanation being the ideal result. As shown in Figure 7, our method accurately highlights the downward spikes, which are fundamental to category judgment. In contrast, baseline methods suffer from issues of redundancy or incompleteness. For instance, TIMEX highlights too many time steps, causing irrelevant information to mix with explanatory temporal patterns, thereby failing to achieve effective interpretation. Conversely, Dynamask omits a significant amount of valid information, failing

to meet the requirement for completeness and resulting in the absence of critical temporal patterns. Similarly, in Figure 8, our method accurately highlights sub-segments with distinct trends. IG and TIMEX++ include excessive redundant background information, while Dynamask and TIMEX miss important trend segments. Figures 9 and 10 demonstrate the performance of the explanation methods on multivariate data. Notably, complex coupling relationships between variables make interpretation more challenging. Our method shows greater adaptability, further validating the advantages of the optimal information retention principle.

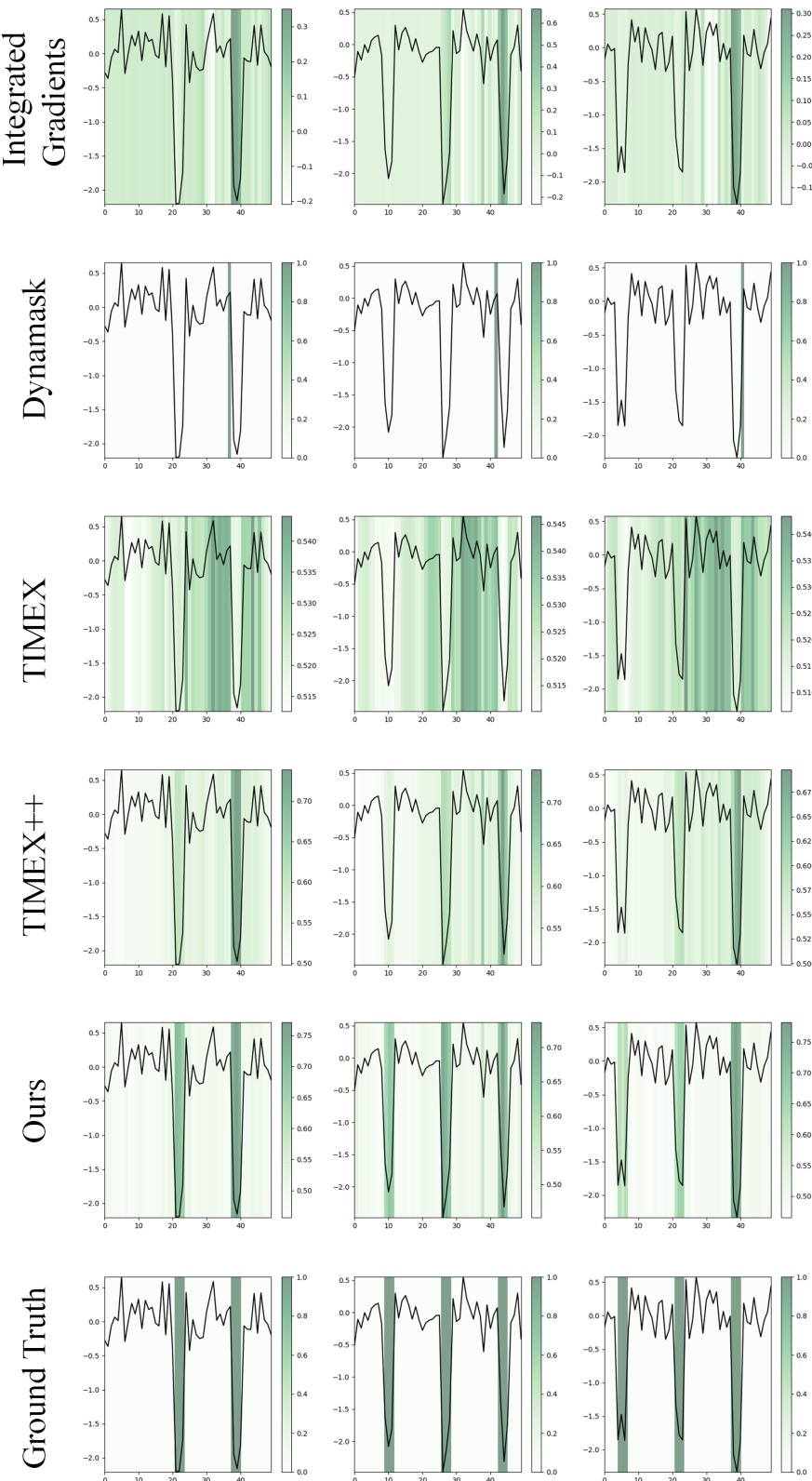

*Figure 7.* Visualization of explanations on FreqShapes dataset.

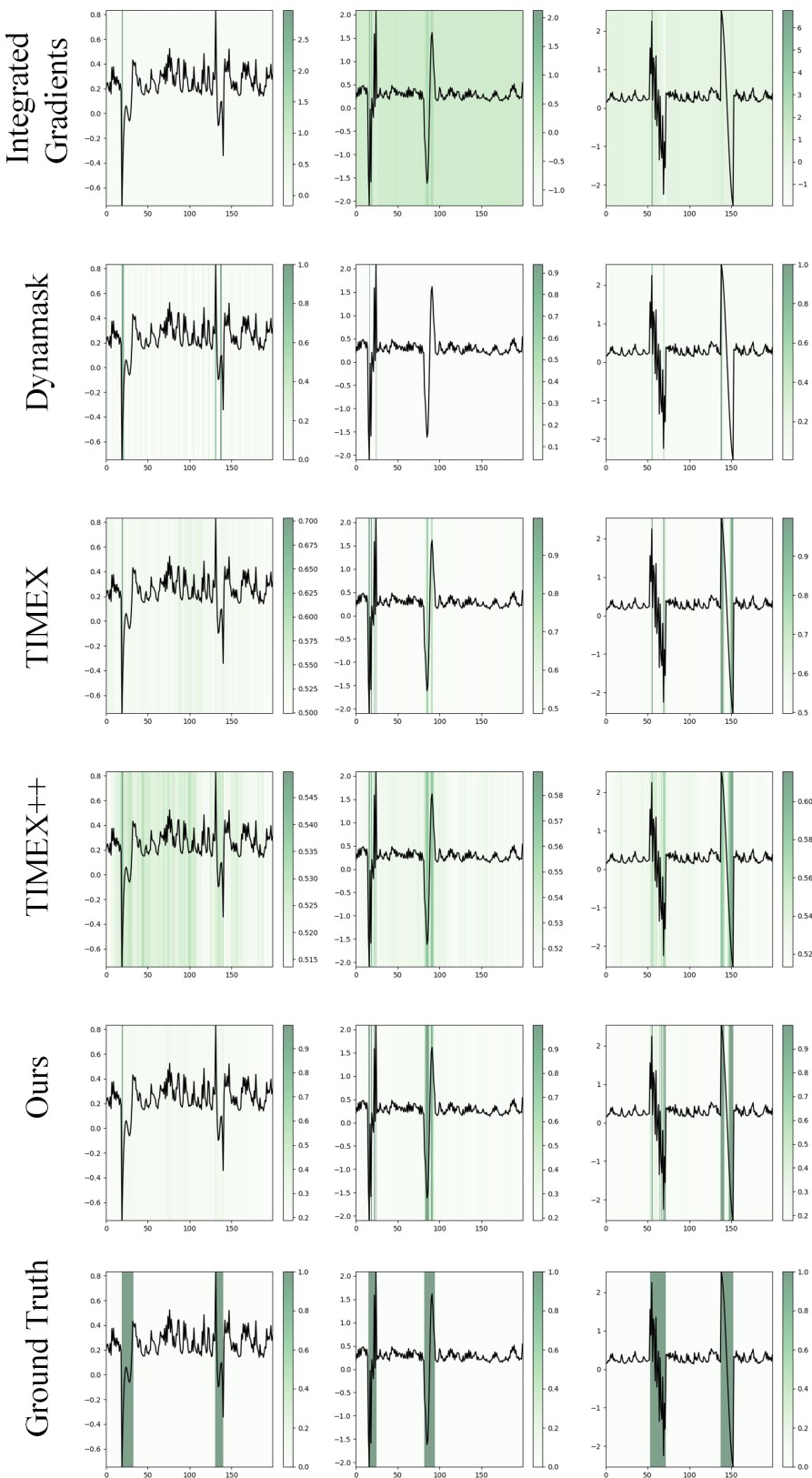

*Figure 8.* Visualization of explanations on SeqComb-UV dataset.

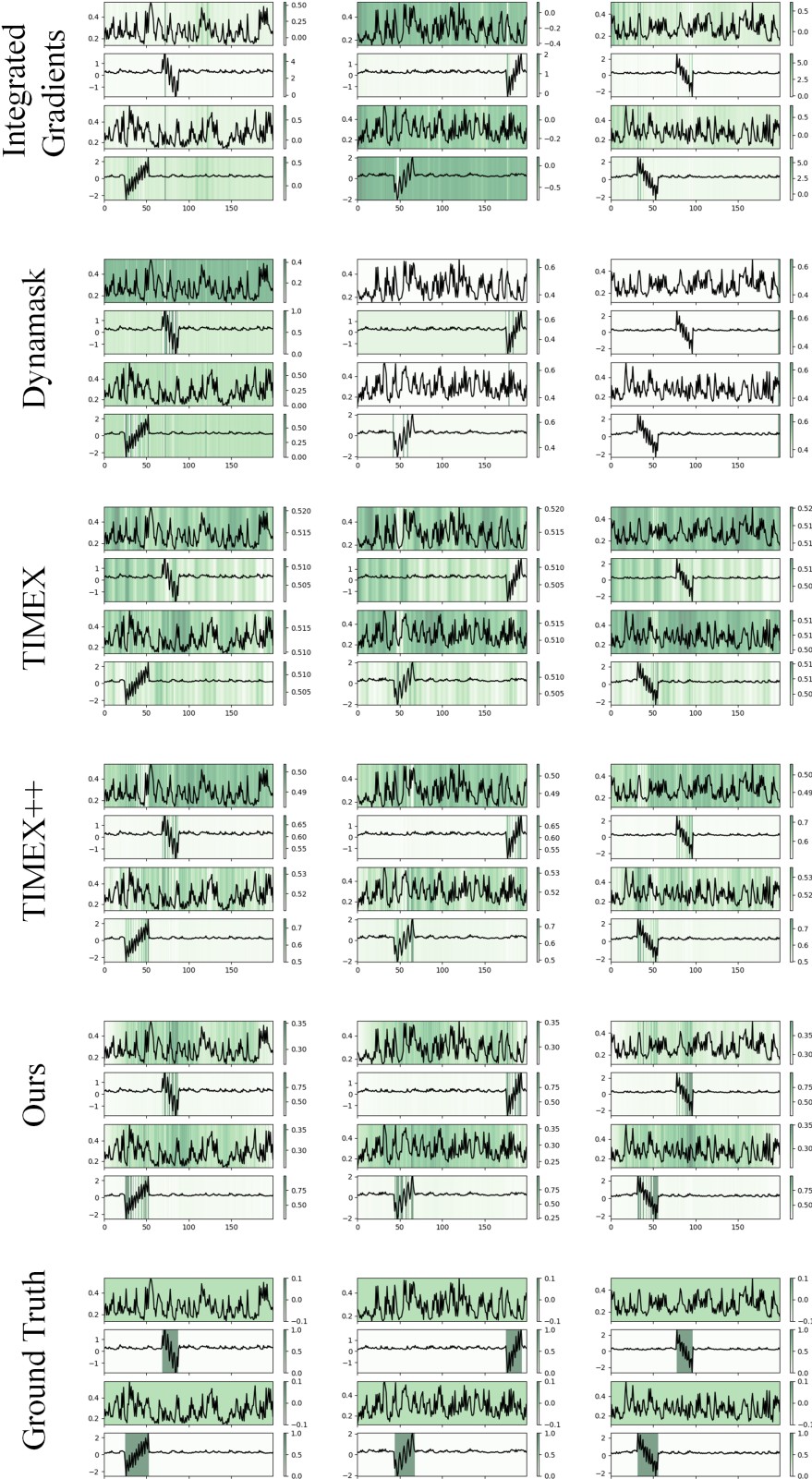

*Figure 9.* Visualization of explanations on SeqComb-MV dataset.

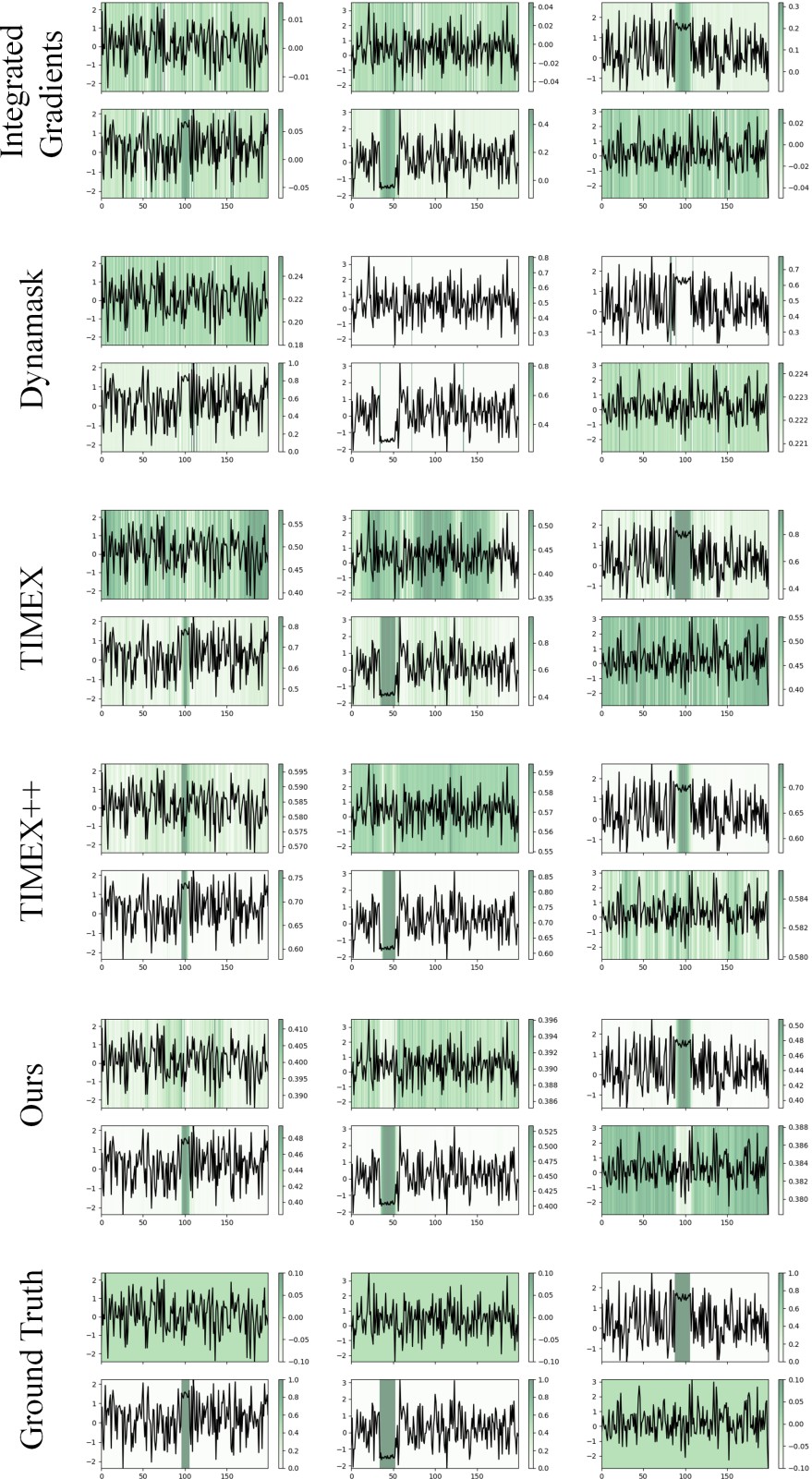

*Figure 10.* Visualization of explanations on LowVar dataset.

