# OpenReview forum: "Optimal Information Retention for Time-Series Explanations"
_ICML.cc/2025/Conference — ICML 2025 poster_

### Official Review · Reviewer_Szo4 · 2025-03-01

**Overall Recommendation:** 4

**Summary:**

This paper proposes the Optimal Information Retention Principle to improve explanations of deep models for time-series data by minimizing redundancy and maximizing completeness using conditional mutual information. The authors develop ORTE, a framework that learns a binary mask to filter irrelevant patterns while preserving key temporal features. ORTE leverages contrastive learning for precise filtering and ensures stable optimization. Experiments demonstrate superior accuracy and completeness over existing methods.

**Claims And Evidence:**

1. This paper employs the Optimal Information Retention Principle to guide the identification of explanatory temporal patterns within time series, including Semantic Information Retention, Minimum Redundant Information Retention, and Maximum Effective Information Retention. In Section 2, three criteria are used to support these claims.

2. The experiments and framework validate the contribution claims made in the introduction.

**Essential References Not Discussed:**

There is no other reference papers from this reviewer.

**Ethical Review Concerns:**

There is no ethical review concerns in this paper.

**Experimental Designs Or Analyses:**

The experimental protocols are well-designed, incorporating quantitative analysis, visualization, and ablation studies. The results and analyses are consistent and well-aligned.

**Methods And Evaluation Criteria:**

1. The method is well-organized and clearly presented, making it easy for readers to follow its main idea. In Figure 2, the Optimal Information Retention Principle is aligned with three components.

2. This paper follows TimeX's evaluation protocol, using four synthetic datasets and four real-world datasets as benchmarks. It employs Area Under Precision (AUP), Area Under Recall (AUR), and Area Under the Precision-Recall Curve (AUPRC) as evaluation metrics. There are no issues with this section.

**Other Comments Or Suggestions:**

Please find the comments above.

**Other Strengths And Weaknesses:**

Strength:
1. This paper introduces a new framework, ORTE, for explainable time series models based on three criteria. The authors validate the effectiveness of these criteria through both theoretical and empirical analyses.
2. The paper is well-organized.

Weakness:
1. In Figure 1, the description is not precise. Since the overlapping area represents mutual information, the green area should correspond to  $H(X)$.
2 . The implementation of Criterion 2.3 is unclear. In line 167, the assumption states: "Assuming $\hat{X}$ has no overlap with the portions of H(X | Y )".   However, the statement, "For the second term, since X and $\hat{X}$ share effective information in $X_m$ , they can naturally regarded as the anchor samples and the positive samples, respectively, in contrastive learning."  contradicts this assumption.

**Questions For Authors:**

Please find the comments above.

**Relation To Broader Scientific Literature:**

This paper proposes the Optimal Information Retention Principle to guide the design of explainable methods in the time series domain and introduces the ORTE framework. Experimental results demonstrate the effectiveness of this approach.

**Theoretical Claims:**

This paper has three criteria: Semantic Information Retention, Minimum Redundant Information Retention, and Maximum Effective Information Retention. In the appendix, the authors provide proofs of the equivalence between these criteria and the implementation loss functions. The proof of Theorem 1 appears to be well-structured, but this reviewer is uncertain about the proof of Theorem 2, particularly the relationship between Equations 32 and 33.

---

> ### Author Rebuttal · Authors · 2025-04-01
>
> **Comment:**
>
> We extend our sincere appreciation to Reviewer Szo4 for providing valuable feedback and acknowledgment of our research.
>
> - **Theoretical Claims:** This paper has three criteria: Semantic Information Retention, Minimum Redundant Information Retention, and Maximum Effective Information Retention. In the appendix, the authors provide proofs of the equivalence between these criteria and the implementation loss functions. The proof of Theorem 1 appears to be well-structured, but this reviewer is uncertain about the proof of Theorem 2, particularly the relationship between Equations 32 and 33.
>
>   **Reply:** For Equations 32 to 33, it is described how to derive positive and negative samples and the corresponding loss function in contrastive learning. Specifically, we learn a mask matrix $M$ that can separate the effective features, and mask $X$ to get $X^m$. However, $X^m$ cannot be directly used as a positive sample, as this would introduce too many 0 values and cause OOD. To do this, we use an MLP to map $X^m$ to $\widehat{X}$. Conversely, the inverse mask filters out redundant features, which can be used as negative samples that are not relevant to the task. The positive sample $\widehat{X}$ and the negative sample $\widehat{X}^-$ thus constructed can be used for contrastive loss, namly, Equations 33.
>
> - **Other Strengths And Weaknesses:**
>
>   - In Figure 1, the description is not precise. Since the overlapping area represents mutual information, the green area should correspond to $H(X)$.
>
>     **Reply:** Thanks for your advice. We will fix this description in an updated version.
>
>   - The implementation of Criterion 2.3 is unclear. In line 167, the assumption states: "Assuming $\widehat{X}$ has no overlap with the portions of $H(X | Y )$". However, the statement, "For the second term, since X and $\widehat{X}$ share effective information in $X_m$ , they can naturally regarded as the anchor samples and the positive samples, respectively, in contrastive learning." contradicts this assumption.
>
>     **Reply:** For $\widehat{X}$ can be considered as the product of $X^m$ learned by MLP, while $X^m$ is considered as the valid feature retained after occlusion. Ideally, $H(X^m)$ should correspond to $I(X; Y)$ consistent. When the information of $Y$ is excluded, the remaining redundant information of $X$ can be considered as irrelevant to $\widehat{X}$, so "no overlap with the portions of $H(X | Y)$". Furthermore, $\widehat{X}$and $X^m$ have information consistency with respect to the label $Y$ that $X$ is oriented to, which is also the motivation for constructing positive samples. So $\widehat{X}$ can be used for contrastive learning from $X^m$. Accordingly, $X$ can be regarded as the anchor of contrastive learning.

---

> > ### Comment · Reviewer_Szo4 · 2025-04-05
> >
> > Thanks for the rebuttal. I would keep the score.

---

> > > ### Author Response · Authors · 2025-04-05
> > >
> > > Thank you for your time and positive comments!

---

### Official Review · Reviewer_w93i · 2025-03-08

**Overall Recommendation:** 2

**Summary:**

This paper proposes an explanation method called "ORTE" (the title of the paper) which uses the Optimal Information Retention principle to construct explanations for time series models. Specifically, given a time series classifier, they propose a mask generating model to produce a binary mask for each input, such that three important information theoretic conditions (Criteria 2.1 - 2.3) hold. These criteria essentially require (1) that the masked input be sufficient for prediction; (2) the masked input has the lowest possible mutual information with the full input; and (3) the complement of the masked input must contain as little information about the label as possible. Experiments demonstrate that the proposed explanations are close to the ground truth explanations, handily beating other methods in this area.

**Claims And Evidence:**

The claims made in the paper are mostly well supported by experiments. However the claim that " We achieve state-of-the-art performance on eight synthetic and real-world time series datasets compared to the latest competitive baselines" is a bit misleading: as the method performs on-par with others (Timex++) on experiments on real-world datasets (Figure 3).

**Essential References Not Discussed:**

The paper misses discussion of key literature in the broader explainability literature that performs masking in a manner similar to this paper. For example:

1. Ruth Fong et al. "Understanding deep networks via extremal perturbations and smooth masks". CVPR 2019
2. Dabkowski and  Gal. "Real time image saliency for black box classifiers".NeurIPS 2017
3. Chen et al, "Learning to explain: An information-theoretic perspective on model interpretation", ICML 2018
4. Yoon et al, "Invase: Instance-wise variable selection using neural networks", ICLR 2019
5. Jethani et al., "Have we learned to explain?: How interpretability methods can learn to encode predictions in their interpretations", AISTATS 2021
6. Bhalla et al, "Discriminative Feature Attributions: Bridging Post Hoc Explainability and Inherent Interpretability", NeurIPS 2023

All of these papers introduce methods for generating per-sample masks, and include discussions of the sufficiency, sparsity criteria similar to this paper. The missing discussion, and the lack of contextualization of the current methods against this larger literature is another critical drawback of this paper.

**Experimental Designs Or Analyses:**

The experimental design is sound and valid.

**Methods And Evaluation Criteria:**

Yes, the proposed methods and evaluation criteria are standard and makes sense for the problem at hand. However, the paper misses a discussion of the interpretability of the masks: while the proposed masks are broadly faithful to the underlying model, it is unclear whether they are also interpretable or meaningful to a human domain expert. It would be good to know for example, if using this method results in greater insights than using other competing methods (e.g.: timex++)? A lack of discussion / experiments / visualizations on this aspect is a critical drawback of this paper.

**Other Comments Or Suggestions:**

N/A

**Other Strengths And Weaknesses:**

Additional Weaknesses:
- The paper introduces 5 criteria in total (criteria 2.1-2.3) + mask sparsity + mask continuity, where the latter two are described more as implementation details. It is unclear which of these criteria is sufficient for good empirical performance in the results shown. For example, what are the trade-offs of the mask continuity and sparsity constraints on the results? The paper lacks a discussion of these aspects.

Additional Strengths:
- The paper demonstrates good empirical performance, especially on recovering known explanations in synthetic settings.

**Questions For Authors:**

Could the authors please clarify the concerns raised regarding:
1. the human-interpretability of the mask explanations, and the impact of the "mask continuity" parameter?
2. the incomplete literature survey, and how the proposed method compares to the literature on masking explanations?

**Relation To Broader Scientific Literature:**

This paper adds onto the literature on time-series explainability using feature attribution, and proposes a method to improve explanation faithfulness. In particular, this improves upon another SOTA time-series explanation method, called timex++, but augmenting the mask generation procedure to include more constraints, such as the minimality of the information between the masked and the full input; and the sparsity of the masks.

**Theoretical Claims:**

I did not check the correctness of the proofs, as they are in the supplementary material.

---

> ### Author Rebuttal · Authors · 2025-04-01
>
> **Comment:**
>
> We express our sincere gratitude to Reviewer w93i for providing comprehensive review, insightful perspectives, and thought-provoking questions.
>
> - **Claims And Evidence:** However the claim that " We achieve ... on real-world datasets (Figure 3).
>
>   **Reply:** Thank you for your advice. We will revise the presentation to faithfully reflect the experimental conclusions.
>
> - **Methods And Evaluation Criteria:** However, the paper misses a discussion of the interpretability of the masks... A lack of discussion / experiments / visualizations on this aspect is a critical drawback of this paper.
>
>   **Reply:** Thank you again for your constructive suggestions, which will be very helpful for a discussion on interpretation and expert cognition. We evaluate the meaning of mask-based explanations with expert explanations from quantitative and qualitative perspectives, respectively.
>
>   1) For synthetic data, we followed the same experimental setup as **TIMEX** and **TIMEX++** with pre-defined Ground-Truth explanations. Quantitative analysis of the consistency between salient features and Ground-Truth can accurately evaluate the interpretation effect. For visualization of synthetic data FreqShapes explanation, as shown in Figure R4 (https://anonymous.4open.science/r/orte/anonymous.pdf). Compared with the strongest baseline **TIMEX++**, our method maintains stronger consistency with the Ground-Truth. Likewise, we provided corresponding visualizations and discussions in the supplementary material of the manuscript.
>   2)  For annotated real-world data, ECG data has expert-predefined Ground-Truth to refer to. Table 2 of the manuscript provides a quantitative evaluation. In order to further explore explain the results and human domain expert cognitive consistency, we visualize our method and the interpretation of the baseline method, as shown in Figure R3 (https://anonymous.4open.science/r/orte/anonymous.pdf). The Figure R3 compares our method with **IG**, **TIMEX++**, and **Ground Truth** (QRS complex waves).  In addition, we invited clinicians to analyze the interpretation results. The professional physician proposed that the Ground Truth in the figure may be mixed with the P wave on the left that does not belong to the QRS wave, and the more accurate Ground Truth should be more narrow, which further verified that our method may provide an insight for the interpretation of real data.
>   3) For the real-world data without annotations, on the one hand, we followed TIMEX and TIMEX++, progressively occluding the saliency features at the bottom of the $p$-percentage. Ideally, the redundant features are assigned low saliency, and all the important features (i.e., completeness) are assigned high saliency. When the low saliency features are occluded, the prediction performance of the algorithm will be less affected. On the other hand, we followed the suggestion of reviewer QJfM and supplemented the imputation experiment to support the requirements of redundancy and completeness in the interpretation results, as detailed in the reply to the Claims And Evidence section of reviewer QJfM.
>
> - **Other Strengths And Weaknesses:** the trade-offs of the mask continuity.
>
>   **Reply:**
>
>   1) For mask continuity, we follow the TIMEX experimental setup, we also investigated the influence of continuity, as shown in Figure R2 (c) (https://anonymous.4open.science/r/orte/anonymous.pdf). We tested AUPRC, AUP, and AUR on SeqComb-MV data. The results showed that continuity did not affect the algorithm significantly, so we chose a consistent setting.
>
>   2) For the sparse mask, Figure R2 (d) (https://anonymous.4open.science/r/orte/anonymous.pdf) investigated the sensitivity. In the interval $[0.001, 0.05]$, the interpretation of the algorithm is relatively stable. However, there will be mode collapse when the sparsity is too large, which is consistent with our expectation that effective features will be missed when the sparsity is too large.
>
> - **Questions For Authors:**
>
>   - Q1. the human-interpretability of the mask explanations, and the impact of the "mask continuity" parameter?
>
>     **Reply:**
>
>     1. For the **human-interpretability of the mask explanations**, please refer to the reply of **Methods And Evaluation Criteria**.
>     2. For the **impact of the "mask continuity" parameter**, please refer to the reply of **Other Strengths And Weaknesses**.
>
>   - Q2. the incomplete literature survey, and how the proposed method compares to the literature on masking explanations?
>
>     **Reply:** We will further clarify our literature survey, and **TIMEX** and **TIMEX++** as our baselines, following masking explanations, both are based on mask learning saliency distributions. And the other baseline method **Dynamask**, is also based on mask learning. In the updated version we will add a more extensive discussion, including time series and other general methods as you mentioned in Essential References Not Discussed.

---

> > ### Comment · Reviewer_w93i · 2025-04-06
> >
> > Thank you for your rebuttal and, for the additional experiments!
> >
> > - **Mask interpretability**: Regarding "progressively occluding the saliency features", perhaps the authors have conflated **interpretability** and **faithfulness** of masks. While interpretability involves assessing whether the masks communicate the rationale for model outputs in human-understandable terms; faithfulness is about assessing whether the provided explanation is consistent with model behaviour. Note that it is possible for masks to be interpretable but not faithful; and also faithful but not interpretable. Also please note that "occluding saliency features" measures faithfulness and not interpretability. Having said that, the visualizations R3 and R4 are helpful to present, and provide an indication that the method may provide human interpretable masks.
> >
> > - **Incomplete Literature survey**: The rebuttal has unfortunately not addressed the central issue, that masking-based methods are common in the literature; dating back to at least 2017 (Dabkowski & Gal). It is unfortunately still unclear how the proposed method compares to this existing literature, and whether it can be interpreted as a straightforward application of these methods to time-series data.
> >
> > For the latter reason, I cannot increase my score at this time.

---

> > > ### Author Response · Authors · 2025-04-08
> > >
> > > We appreciate your thorough review and valuable suggestions. We particularly value your insightful analysis regarding interpretability and faithfulness, as well as your positive feedback on visualization R3 and R4. Regarding your latter concern about **Incomplete Literature survey**, we respond as follows:
> > >
> > > Mask-based methods are devoted to learning a mask matrix $M$, and the Hadamard products of the input $ X $ and $M$ are represented as the salient features. [1] proposed to learn a masking model by embedding category conditions into the U-Net structure to realize the saliency detection. The sparsity and smoothness terms optimize the objective function of mask learning, similar to that in our paper. [2] maximizes the mutual information between the selected features and the conditional distribution of the response variables from an information theoretic perspective, and develops a variational approximation to estimate this mutual information. Although it can effectively filter redundant information, it may cause the omission of important features, which is the completeness claimed in our manuscript. INVAS[3], inspired by the actor-critic methodology, proposes a selector network for mask learning, optimized in combination with a prediction network and a baseline network. Extreme perturbations[4], which are perturbations that have the largest impact on the network among all perturbations within a fixed region, are used to trade off the optimization term of the mask constraint. Moreover, this perturbation can be extended to intermediate activation layers to explore diverse properties of feature representations. [5] points out that explanation methods suffer from computational efficiency, inaccuracy, or lack of faithfulness. The lack of robustness of the underlying black-box models, especially to the erasure of unimportant distractor features in the input is a key reason why certain attributions lack faithfulness[6]. The Distractor Erasure Tuning method [6] is proposed that adapts black-box models to be robust to distractor erasure,  thus providing discriminative and faithful attributions. Other interpretive methods specifically designed for time series have also received extensive attention in recent years. For example, Dynamask [7] considers the time dependence and learns the influence of dynamic perturbation operators on the mask. TIMEX [8] specifically designs surrogate models in order to avoid the inductive bias of other methods on time series data, and identifies explanation temporal patterns by aligning latent space feature consistency and predictive distribution consistency. As an advanced version of TIMEX, TIMEX++ [9] alleviates the trivial solutions and distribution shift based on information bottleneck.
> > >
> > > The above approaches provide practical solutions for mask-based explanations from different perspectives, including but not limited to heuristics, information theory, etc. However, there is still a lack of comprehensive consideration of information redundancy and completeness, which is easy to cause the mixing of invalid information or the lack of effective information. On the other hand, differences in data modalities may cause inductive bias. To this end, we propose the optimal information retention principle and derive the corresponding objective functions. Similar to [1], we adopt the sparsity and smoothness constraints on the mask. The learning of a Bernoulli distribution can provide an explanation probability basis for the mask, and contrastive learning is used to separate effective features from redundant features. We propose the ORTE method as a practical solution for time series data, which can be adapted to various time series models. In addition, we propose adapt-STE to decouple the discrete mapping process and alleviate the differentiable limitation.
> > >
> > > 1. Dabkowski and Gal. "Real time image saliency for black box classifiers".NeurIPS 2017
> > > 2. Chen et al, "Learning to explain: An information-theoretic perspective on model interpretation", ICML 2018
> > > 3. Yoon et al, "Invase: Instance-wise variable selection using neural networks", ICLR 2019
> > > 4. Ruth Fong et al. "Understanding deep networks via extremal perturbations and smooth masks". CVPR 2019
> > > 5. Jethani et al, "Have we learned to explain?: How interpretability methods can learn to encode predictions in their interpretations", AISTATS 2021
> > > 6. Bhalla et al, "Discriminative Feature Attributions: Bridging Post Hoc Explainability and Inherent Interpretability", NeurIPS 2023
> > > 7. Crabb´e, J et al,  Explaining time series predictions with dynamic masks, ICML 2021
> > > 8.  Queen et al, Encoding time-series explanations through self-supervised model behavior consistency. NeurIPS 2024
> > > 9. Liu, Z et al, Timex++:Learning time-series explanations with information bottleneck. ICML 2024

---

### Official Review · Reviewer_voUp · 2025-03-13

**Overall Recommendation:** 4

**Summary:**

This paper introduces a novel approach to explainability in time-series deep learning models through an information-theoretic lens. The authors propose the "Optimal Information Retention Principle" which outlines three key criteria for high-quality explanations via information retention: ‘semantic’, ‘minimum redundant’, and ‘maximum effective’. Based on these they develop ORTE, a practical framework that learns binary masks to identify important patterns while filtering out redundant information. The framework incorporates contrastive learning to achieve a balance between low redundancy and high completeness in explanations. Experiments on both synthetic and real-world datasets demonstrate that ORTE outperforms existing SOTA explainability methods on similar metrics to those used by TIMEX/TIMEX++.

**Claims And Evidence:**

Overall, the core technical claims about ORTE's performance improvements are well-supported, but some broader claims about its practical advantages and generalizability would benefit from additional evidence.

Well-supported:
C1. Performance superiority: The claim that ORTE outperforms existing methods is convincingly supported by comprehensive experiments across multiple datasets. The quantitative results in Tables 1-2 show clear improvements over competitive baselines, particularly on AUPRC, AUP, and AUR metrics.
C2. Theoretical foundation: The claim that information theory can provide optimization criteria for explanations is supported through detailed mathematical derivations and proofs in the paper and appendices.
C3. Adaptability: The claim that ORTE works across different model architectures is demonstrated through additional experiments on CNNs and LSTMs in the appendix.

Less convincing evidence:
C4. Practical utility: claim lacks detailed comparison of computational demands versus simpler approaches, which would be important for real-world applications. Introducing (even theoretical) estimation of time/memory complexity would be welcome.
C5. Real-world evaluations: while the occlusion experiments on real-world datasets without ground truth (Figure 3) show ORTE maintains high prediction AUROC, this is only a proxy measure and doesn't definitively prove the explanations are correctly identifying the truly important features. Other work has introduced evaluations by consulting subject-matter experts and comparing their interpretation of variables of importance with those generated by the proposed framework.

**Essential References Not Discussed:**

The work does address most relevant work, however it excludes recent advances on applying multiple instance learning to time series tasks. Early et. al. (2024) [https://arxiv.org/pdf/2311.10049] and Chen et. al. (2024) [https://arxiv.org/abs/2405.03140] provide an introduction to these methods and how they provide inherent interpretability for multivariate time series classification problems from an information-theoretic perspective.

Earlier work by Tonekebani et. al. (2020) [https://arxiv.org/abs/2003.02821]  and Ismail et. al. (2020) [https://arxiv.org/abs/2010.13924] which helped set the stage for time series interpretability and provides other synthetic and real-world evaluations for improved experimentation could further support this work.

**Experimental Designs Or Analyses:**

Again, the synthetic experimental design is comprehensive and appropriate:
- The use of synthetic datasets with ground truth explanations (FreqShapes, SeqComb-UV, SeqComb-MV, LowVar) provides a clear baseline for evaluating explanation quality.
- The ablation studies effectively isolate the contribution of each component in the framework.
- The visualizations provide qualitative evidence that complements the quantitative metrics.

However, the experimental design could be improved:
- While  real-world datasets from different domains were included (ECG, PAM, Epilepsy, Boiler) the experiments themselves were not rigorously compared.
- The authors could provide runtime/computational complexity comparisons across methods
- A hyperparameter sensitivity analysis could be completed

**Methods And Evaluation Criteria:**

The proposed method of using binary masks to identify important regions via a transformer-based mask generator -> contrastive separator -> predictive distribution aligner makes sense after the introduction of the three information retention criteria.

The synthetic evaluations with ground truth are standard for time series interpretability and show the performance of the model against a wide range of competing methods. Although the real-world evaluation is somewhat lacking, the approach does not seem too far off. There is some evaluation missing when considering how features are being masked - it seems the authors are simply masking the bottom x-percentile but this does not take into account temporal dependencies within each dimension of a multivariate time-series - they should consider masking until a 1-std change or until the end of the series.

**Other Comments Or Suggestions:**

N/A

**Other Strengths And Weaknesses:**

S1. The paper provides a unique and theoretically sound unified optimization framework for TS explainability
S2. The adapt-STE technique is a novel contribution that could be useful for other binary mask learning problems.
S3. The synthetic evaluations show significant performance improvements compared with previous SOTA.

W1. The real-world evaluations as previously mentioned.
W2. No significant investigation into edge-cases like dealing with irregularly sampled or missing data.

**Questions For Authors:**

Q1. How does the computational complexity of ORTE compare to methods like IG or TIMEX++?
Q2. How sensitive is the method to the choice of hyperparameters, particularly those in the contrastive learning component?

**Relation To Broader Scientific Literature:**

The paper situates itself well within existing TS explainability literature, particularly at the intersection of time-series analysis and information theory. The authors appropriately acknowledge:
- Prior work on local explanations in time series (Dynamask, TIMEX, WinIT)
- Information-theoretic approaches in deep learning (Information Bottleneck, contrastive learning)
- The challenges specific to time-series explainability (temporal dependencies, out-of-distribution issues)
The work advances the field by providing a unified information-theoretic framework specifically designed for time-series data, addressing the limitations of methods that were originally designed for images or text.

**Theoretical Claims:**

I reviewed the proofs provided in Appendices B.1 and B.2, which derive the objective functions from the information-theoretic principles. The mathematical derivations appear sound and follow established information theory principles.
No errors were identified in these derivations, and the theoretical foundation appears solid.

---

> ### Author Rebuttal · Authors · 2025-04-01
>
> **Comment:**
>
> We sincerely appreciate Reviewer voUp for offering valuable insights and recognizing of our work.
>
> - **Claims And Evidence:**
>
>   - C4. Practical utility: claim lacks detailed comparison of computational demands versus simpler approaches, which would be important for real-world applications. Introducing (even theoretical) estimation of time/memory complexity would be welcome.
>
>     **Reply:** We evaluated the computational requirements of interpretability methods on two real-world datasets (PAM and Epilepsy), as detailed in Table R1 (https://anonymous.4open.science/r/orte/anonymous.pdf). The computing infrastructure consisted of Ubuntu 18.04.6 LTS and an NVIDIA GeForce RTX 2080. Table R1 compares our method (**ORTE**) with four baseline approaches (**IG**, **Dynamask**, **TIMEX**, and **TIMEX++**) in terms of parameters (M), FLOPs (G), and Inference Runtime (s). Compared with **TIMEX** and **Timex++**, our inference time is slightly longer due to the *double*-STE to implement the regulation of discrete maps, but all three are in the same rank, i.e., less than 1 second.  **IG** and **Dynamask** require longer inference time, which is consistent with expectations since both methods operate recursively on a single sample, while our method requires only one forward propagation.
>
>   - C5. Real-world evaluations: while the occlusion experiments on real-world datasets without ground truth (Figure 3) show ORTE maintains high prediction AUROC, this is only a proxy measure and doesn't definitively prove the explanations are correctly identifying the truly important features. Other work has introduced evaluations by consulting subject-matter experts and comparing their interpretation of variables of importance with those generated by the proposed framework.
>
>     **Reply:** Thanks for your advice. We consulted clinical experts and analyzed real-world **ECG** data, as shown in Figure R3 (https://anonymous.4open.science/r/orte/anonymous.pdf). Figure R3 compares our method with **IG**, **TIMEX++**, and **Ground Truth** (QRS complex waves). Although **IG** accurately locates some points of the QRS, more information is missed compared with the Ground-Truth. While **TIMEX++** highlights the salient features, it also mixes the redundant features, which makes it difficult to understand the prediction of the algorithm. Our method maintains less redundancy while highlighting more complete salient features, which verifies the optimal information retention principle proposed in this paper. Notably, clinical experts identified a critical annotation discrepancy: the reference Ground Truth inadvertently incorporated proximal P-wave components (non-QRS elements) at leftward leads. This observation suggests that the benchmarks may require narrower physiological bounds, which further supports that our method may provide new insight for the interpretation of real data.
>
> - **Other Strengths And Weaknesses:**
>
>   - W1. The real-world evaluations as previously mentioned.
>
>     **Reply:** For the evaluation of real-world data, on the one hand, we followed your friendly advice and consulted clinical experts as a supplement to the explanation evaluation, as mentioned in C5 reply of **Claims And Evidence**. On the other hand, we followed the advice of reviewer QJfM and supplemented the imputation experiments, as detailed in the reply to the Claims And Evidence section of reviewer QJfM.
>
>   - W2. No significant investigation into edge-cases like dealing with irregularly sampled or missing data.
>
>     **Reply:** We investigated edge-case of missing data, as shown in Table R2 (https://anonymous.4open.science/r/orte/anonymous.pdf). We compare the proposed method **ORTE** with the strongest baseline **TIMEX++** on SeqComb-MV data. We set three scenarios including random loss of 5% and 15% of data points and random loss one variable, respectively. The results show that our method performs well on AUPRC, AUP and AUR, which is consistent with the claims of the manuscript.
>
> - **Questions For Authors:**
>
>   - Q1. How does the computational complexity of ORTE compare to methods like IG or TIMEX++?
>
>     **Reply:** Please refer to C4 reply of **Claims And Evidence**.
>
>   - Q2. How sensitive is the method to the choice of hyperparameters, particularly those in the contrastive learning component?
>
>      **Reply:** We investigated the hyperparameter choice of contrastive learning (i.e., $ \alpha $), as shown in Figure R2(b) (https://anonymous.4open.science/r/orte/anonymous.pdf).  When $0.1 \leqslant \alpha \leqslant 13 $, AUPRC and AUP slowly rise and gradually plateau, while AUR gradually rises. The AUP will decrease significantly when $\alpha $ is larger, that is, the explanation fails. A value proximate to 10 is recommended for our experimental applications.
>
> - **Essential References Not Discussed:**
>
>   **Reply:** Thanks. We will supplement and analyze these references in the updated version.

---

> > ### Comment · Reviewer_voUp · 2025-04-02
> >
> > Thanks to the authors for responding. After reviewing the updated responses including the expansion of the computational complexity, the hyperparameter reviews, and the additional experimental data I have further confidence in my current rating which I will keep.

---

> > > ### Author Response · Authors · 2025-04-02
> > >
> > > Thank you once again for your time and valuable suggestions. We will include the above discussions in the revised version.

---

### Official Review · Reviewer_QJfM · 2025-03-16

**Overall Recommendation:** 4

**Summary:**

The authors address redundancy and completeness in time-series explanation methods by deriving an Optimal Information Retention principle from information theory, which optimizes explanations by minimizing redundancy while maximizing completeness. Based on this principle, they propose ORTE, a novel explanation framework that ensures informative and non-redundant explanations. They validate their approach through empirical evaluations on eight synthetic datasets and four real-world datasets, demonstrating its effectiveness over existing explanation methods.

**Claims And Evidence:**

The paper proposes ORTE, an explanation framework designed to minimize redundancy and maximize completeness, verified through AURPC, AUP, and AUR on synthetic datasets with ground truth explanation labels. The evaluation methodology for synthetic datasets is well-structured and justified, as these metrics align with the paper’s theoretical objectives. Table 1 quantitatively supports the validity of ORTE in this controlled setting.

For real-world datasets, the paper employs stepwise occlusion experiments (Figure 3) to assess explanation validity. The use of AUROC as a primary metric makes sense in measuring how explanations influence model predictions. However, there are some limitations in the evaluation approach.
1. Occlusion alone does not fully capture redundancy and completeness. The authors assume that a stable AUROC under extreme feature removal suggests explanation stability. However, if AUROC remains above 95% after occluding 99% of features, this may indicate a redundant explanation rather than a robust one.
2. Insertion experiments should complement occlusion. The paper does not include stepwise insertion experiments, which could provide a more balanced assessment. Insertion tests could verify completeness by measuring how much AUROC recovers when adding back features, while also assessing redundancy by identifying how many features are actually necessary.

**Essential References Not Discussed:**

The paper appropriately cites works related to explainability, information-theoretic approaches, and time-series interpretability. Its main motivation—the lack of a unified optimization criterion across explanation methods—is closely related to the evaluation of interpretability techniques. To strengthen this connection, the authors could discuss ROAR ([1], NeurIPS 2019), which introduced a systematic framework for assessing interpretability methods and formalized insertion and occlusion (deletion) as key evaluation techniques. Referencing ROAR would further validate the effectiveness of ORTE's evaluation strategy.
Reference:[1] Hooker, S., Erhan, D., Kindermans, P. J., & Kim, B. (2019). A benchmark for interpretability methods in deep neural networks. Advances in Neural Information Processing Systems, 32.

**Experimental Designs Or Analyses:**

As discussed in the Claims and Evidence section, integrating stepwise insertion experiments alongside occlusion would provide a more comprehensive assessment of redundancy and completeness, addressing current limitations in evaluating explanation quality.
Additionally, a minor potential improvement could be to analyze the sensitivity of the hyperparameter controlling the noise level in negative samples used in contrastive learning. Since the method applies Gaussian noise to generate negative samples, adjusting the noise level could have some influence on the contrastive learning process. While this is not a critical issue, a brief sensitivity analysis could help ensure the robustness of the approach.

**Methods And Evaluation Criteria:**

The validation strategy—conducted on eight synthetic datasets and four real-world datasets—is appropriate for the problem and application. The use of synthetic datasets with ground truth explanations allows for a well-controlled evaluation, and the metrics (AURPC, AUP, AUR) provide a reasonable measure of redundancy and completeness in this setting.
However, for real-world datasets, the experiments are not sufficient to effectively demonstrate low redundancy and high completeness. The evaluation primarily relies on stepwise occlusion experiments, which, while informative, do not fully capture both aspects of explanation quality. As discussed in the Claims and Evidence section, integrating stepwise insertion experiments alongside occlusion would provide a more comprehensive assessment of redundancy and completeness.

**Other Comments Or Suggestions:**

1. Typo. Line 208 (2.3)
2. Hyperlink issue. The hyperlinks for references and sections did not work on my end. This may be a local issue, but please double-check to ensure proper linking.

**Other Strengths And Weaknesses:**

Another strength is clear Motivation and Theoretical Grounding. The paper effectively frames the problem of unifying redundancy minimization and completeness maximization in explainability, providing a principled approach grounded in information theory. Other than Limited Experimental Evaluation for Real-World Datasets and Gaussian Noise Imputation for Negative Samples, there are no concerns.

**Questions For Authors:**

- Comment on Figure 3. Beyond stating that the experimental results are better and more stable, what additional interpretations can be drawn from Figure 3? Specifically, how do the results relate to redundancy and completeness? A more detailed discussion on these aspects would enhance the interpretation of the figure.

**Relation To Broader Scientific Literature:**

The paper builds on prior work in explainability by introducing ORTE, which optimizes redundancy minimization and completeness maximization. This aligns with information-theoretic approaches in explainability and contributes to time-series interpretability. While the Gaussian noise imputation technique for negative samples is not a major contribution, it serves as a simple way to avoid capturing OOD patterns, unlike the zero-padding approach mentioned in the related work section.

**Theoretical Claims:**

The proofs for the two main criteria—minimum redundant retention and maximum effective retention—are provided in the supplementary material. Upon review, they appear to be correct with no immediate inconsistencies.

---

> ### Author Rebuttal · Authors · 2025-04-01
>
> **Comment:**
>
> We sincerely appreciate Reviewer QJfM for the careful analysis of our work and valuable suggestions.
>
> - **Claims And Evidence:** 1. Occlusion alone does not fully capture redundancy and completeness. & 2. Insertion experiments should complement occlusion.
>
>   **Reply:** Thanks again for your advice. We believe the insertion experiments can serve as a beneficial supplement to our occlusion experiments. We completed the insertion experiments by gradually inserting the bottom $p$-percentage salient features, as shown in Figure R1 (https://anonymous.4open.science/r/orte/anonymous.pdf). Similar to the occlusion experiments, the imputation experiments compare our method (**ORTE**) with three baseline methods (**Dynamask**, **TIMEX**, and **TIMEX++**) and random insertion (**Random**) on three real-world datasets (**PAM**, **Epilepsy**, and **Boiler**). The results show that **ORTE** achieves a lower AUPROC when inserting the bottom 75% of salient features. As the insertion percentage increases, the predicted AUROC gradually improves. When the insertion ratio reaches 97.5%, ORTE attains the highest AUPROC. This indicates that the most interpretable or informative features are concentrated in the high-saliency features, further validating the manuscript's claims of low redundancy and high completeness. In addition, we notice that **Random** always maintains a high AUROC, which is because the random selection of features does not distinguish the salient or importance of features and contains informative time points or segments.
>
> - **Other Strengths And Weaknesses:** Other than Limited Experimental Evaluation for Real-World Datasets and Gaussian Noise Imputation for Negative Samples, there are no concerns.
>
>   **Reply:**
>
>   1) As replied in **Claims and Evidence**, the imputation experiments, serving as a complementary evaluation metric on real-world data, is able to jointly support the low redundancy and high completeness claims of this paper together with the occlusion experiments.
>   2) We investigated the impact of Gaussian noise intensity on negative sample constructions, as shown in Figure R2 (a) (https://anonymous.4open.science/r/orte/anonymous.pdf). We tested $ \sigma = [0.1, 0.3, 0.5, 1.0, 1.5, 2.0, 2.5, 3.0] $ eight groups of noise intensities on the multivariate synthetic dataset (**SeqComb-MV**). The results show that AUP is lower at $\sigma \leqslant 1.0 $, while AUPRC and AUR are higher, which indicates that insufficient noise intensity may be not beneficial for the separation of interpretation patterns from redundant information. When $1.5 \leqslant \sigma \leqslant 3.0 $, AUP and AUPRC are higher, which indicates that the interpretation patterns of positive samples are effectively discriminable and do not produce OOD patterns. A value proximate to 2.5 is recommended for our experimental applications. And the more analysis of gaussian noise imputation for negative samples on other datasets will be included in the updated version.
>
> - **Questions For Authors:**  Comment on Figure 3. ... how do the results relate to redundancy and completeness?
>
>   **Reply:** For Figure 3, we stepwise occluded the bottom $p$-percentage salient features, following TIMEX and TIMEX ++. Ideally, the redundant features are assigned low saliency, and all the important features (i.e. completeness) are assigned high saliency. When the low saliency features are occluded, the prediction performance of the algorithm will be less affected. On the contrary, if the important features are identified as low saliency and are occluded, the prediction performance of the algorithm will be significantly reduced. On the one hand, this can reflect the requirements of redundancy and completeness in interpretation results, but also as pointed out in **Claims and Evidence**, due to the lack of ground-truth interpretation of real-world data, occlusion experiments still have the limitation of robustness. To this end, the imputation experiment can serve as a powerful supplement. Specifically, when the low saliency features are occluded or the high saliency features are inserted, the proposed algorithm can obtain higher prediction performance, which will further verify the claim of this paper.
>
> - **Essential References Not Discussed:** To strengthen this connection, the authors could discuss ROAR ([1], NeurIPS 2019) ... Referencing ROAR would further validate the effectiveness of ORTE's evaluation strategy.
>
>   **Reply:** Thanks for the friendly suggestion, and we will discuss the reference of ROAR in the updated version.
>
> - **Other Comments Or Suggestions:** 1. Typo. Line 208 (2.3) & 2. Hyperlink issue.
>
>   **Reply:**
>
>   1) We addressed this issue in the updated version.
>   2) We double-checked the hyperlinks to references and sections on multiple reading platforms.

---

> > ### Comment · Reviewer_QJfM · 2025-04-07
> >
> > Thank you for adding the insertion experiment and hyperparameter sensitivity analysis. This additional evaluation complements the existing occlusion experiment and allows for a more balanced assessment.
> > However, I believe that the current occlusion and insertion experiments still fall short of directly demonstrating low redundancy and high completeness. Specifically, since the experiments are conducted based on bottom-k% salient features—removing or adding the least salient features first—they only allow for an indirect evaluation of completeness.
> > For a more direct assessment, it would be more appropriate to perform by adding top-k% salient features first, which would reveal how quickly the model performance recovers when only the most informative features are used. For an explanation to be low redundancy, the model performance should change significantly when either low- or high-saliency features are removed or added. This is because, when information is concentrated and non-redundant, small perturbations to the key features will lead to sharp performance changes.
> > In contrast, high completeness implies that the top salient features alone should be sufficient to recover the model’s original performance. That is, the explanation must contain the majority of the essential information. Since the current experiments are based on bottom-k% insertion, we can evaluate completeness indirectly by observing how much the model’s performance recovers in the end.
> > From this perspective, the proposed method demonstrates the highest recovery performance (excluding the random baseline) in the insertion experiment, which suggests that it captures more informative features than the other methods. Furthermore, the area under the performance curve in the occlusion experiment is comparable to or slightly better than the baselines, which is also a positive signal.
> > Based on these improvements and observations, I am updating my score (3->4).

---

> > > ### Author Response · Authors · 2025-04-07
> > >
> > > Thanks for your valuable suggestions and positive comments!

---

### Decision · Program_Chairs · 2025-05-01

**Decision:**

Accept (poster)

**Comment:**

The authors propose an approach based on the optimal information retention principle from information theory to address redundancy and completeness in time-series explanation. This approach minimizes redundancy while maximizing completeness when generating explanations.

The reviewers appreciated the work and the "clear motivation" of the work and its "theoretical grounding". While the reviewers generally liked the experimental design, they recommended additional sensitivity analysis (see Reviewer QJfM's response to the rebuttal). I think including such an analysis would be insightful. The reviewers have also pointed out missing references to recent advances on applying multiple instance learning to time series tasks and broader explainability literature that performs masking. I think the authors should expand the related work to include these works.